# A new root-knot nematode, *Meloidogyne vitis* sp. nov. (Nematoda: Meloidogynidae), parasitizing grape in Yunnan

**Yanmei Yang**[1,2], **Xianqi Hu**[1,2]*, **Pei Liu**[1,2], **Li Chen**[3], **Huan Peng**[4], **Qiaomei Wang**[1,2], **Qi Zhang**[1,2]

**1** College of Plant Protection, Yunnan Agricultural University, Kunming, Yunnan Province, China, **2** State Key Laboratory for Conservation and Utilization of Bio-Resources in Yunnan, Yunnan Agricultural University, Kunming, Yunnan Province, China, **3** Wheat Research Institute, Shanxi Academy of Agricultural Sciences, Linfen, Shanxi Province, China, **4** State Key Laboratory for Biology of Plant Disease and Insect Pests, Institute of Plant Protection, Chinese Academy of Agricultural Science, Beijing, China

* xqh@ynau.edu.cn

## Abstract

An unknown root-knot nematode was found at high density on grape roots collected from Yunnan Province. Morphometric traits and measurements, isozyme phenotypes, and molecular analysis clearly differentiated this nematode from previously described root-knot nematodes. This new species is described, illustrated and named *Meloidogyne vitis* sp. nov. The new species can be distinguished from other *Meloidogyne* spp. by a unique combination of characters. Females display a prominent neck, an excretory pore is located on the ventral region between 23rd and 25th annule behind lips, an EP/ST ratio of approximately 2.5 (1.98–2.96), a perineal pattern with two large and prominent phasmids, and a labial disc fused with the medial lips to form a dumbbell-shaped structure. Males display an obvious head region, a labial disc fused with the medial lips to form a dumbbell-shaped structure, no lateral lips, a prominent slit-like opening between the labial disc and medial lips, a distinct sunken appearance of the middle of the medial lips, and four incisures in the lateral field. Second-stage juveniles are characterized by a head region with slightly wrinkled mark, a labial disc fused with the medial lips to form a dumbbell-shaped structure, a slightly sunken appearance of the middle of the medial lips, a slit-like amphidial openings between the labial disc and lateral lips, and four incisures in the lateral field. The new species has rare Mdh (N3d) and Est phenotypes (VF1). Phylogenetic analysis based on ITS1-5.8S-ITS2, D2D3 fragments of rDNA, and coxI and coxII fragments of mtDNA sequences clearly separated the new species from other root-knot nematodes, and the closest relative was *Meloidogyne mali*. *Meloidogyne mali* was collected for amplifying these sequences as mentioned above, which were compared with the corresponding sequences of new species, the result showed that all of these sequences with highly base divergence (48–210 base divergence). Moreover, sequence characterized amplified region (SCAR) primers for rapid identification of this new species were designed.

**Data Availability Statement:** All sequences are available from the GenBank database (accession number: MN816222, MN816223, MN816224, MN816225, MN816226, MN814829, MN814830,

MN814831, MT012386 and MT012387). Other relevant data are within the paper and its Supporting Information files.

**Funding:** This work was supported by grants from the National Key Research and Development Project (Nos. 2018YFD0201202; 2017YFD0200601). The funders had no role in study design, data collection and analysis, decision to publish, or preparation of the manuscript.

**Competing interests:** The authors have declared that no competing interests exist.

## Introduction

Root-knot nematodes (RKNs), belonging to family Heteroderidae, are the most important categories of plant-parasitic nematode and parasitize a large number of plant species [1]. After being parasitized, plants often exhibit severely reduced production, causing significant economic losses. More than 90 species of RKNs have been described to date [2, 3], which are able to infect virtually any species of higher plant and have a near cosmopolitan distribution [4]. In recent years, new RKN species have been found in woody plants such as coffee, olive and kiwifruit [5–7].

*Vitis vinifera* (Vitaceae, *Vitis L.*) is one of the most widely grown fruit crops in many areas of the world [8]. It is a woody vine plant whose fruits can be eaten raw as a fresh fruit or made into dried fruit, juices and wine; thus, it is of great economic value. Grape cultivation is believed to have originated in Armenia, near the Caspian Sea in Russia, from where it spread westward to Europe and eastward to Iran and Afghanistan [9]; at present, it is widely grown in tropical, temperate and subtropical regions worldwide. China is home to the most abundant grape genetic resources in the world, where grape cultivation has been conducted for thousands [10]. Yunnan Province is one of the major provinces for the grape industry in China. As of the end of 2014, the total output value of grape in Yunnan exceeded 7 billion yuan; thus, grape plays an important role in the agricultural industry in this province [11]. However, various pathogens, including plant-parasitic nematodes, pose a serious threat to the production of grapes worldwide, and RKNs are one of the important factors restricting grape production [12, 13].

Root-knot nematode infestation of grape has been documented in southern Australia, South Africa, France, the United States and other countries [14]. Grapesroot system are well developed and is the target of RKN infection, and both seedlings and adult plants can be harmed. The destroyed fibrous roots and root hairs initially show slight swelling. In later stages, the diseased roots become rotten, which directly affects the absorption of water and nutrients by the root system and results in great reductions in grape yield and fruit quality. In Australia, almost all vineyards on sandy soils are infected by RKNs [15], and four species (*Meloidogyne incognita*, *Meloidogyne arenaria*, *Meloidogyne javanica* and *Meloidogyne hapla*) were found to damage grapes in southern Australia [16]. *Meloidogyne incognita* and *M. hapla* are common grape root pests [17], and Liu and Zhang (2017) reported that grapes from the Huaihai economic zone were infected by *M. incognita* [18]. *Meloidogyne javanica* is the predominant RKN in Australian vineyards [12], and *M. hapla* is abundant and widespread in Washington's semiarid vineyards [19]. *Meloidogyne incognita*, *M. arenaria*, *M. javanica* and *M. hapla* are the main species parasitizing grapes [14, 20]. However, three other *Meloidogyne* species, *Meloidogyne nataliei*, *Meloidogyne ethiopica* and *Meloidogyne thamesi*, also infect grapes [21–23]. RKNs can severely reduce grape yield, causing significant economic losses. Li *et al.* (2006) reported that *M. incognita* was the most common RKN in vineyards, where it could reduce the yield of susceptible grape varieties by approximately 80% and that of resistant grape varieties by approximately 40% [24]. Furthermore, RKNs can also interact with bacteria, fungi and other pathogens to further increase damage to grapes.

Given that grapes are seriously damaged by RKNs, the accurate identification of pathogen species could be crucial for designing effective prevention and control strategies. However, in China, there have been few studies on grape RKNs in recent years, with problems such as unclear distributions, unclear species and a lack of prevention and control technologies. In addition, the RKNs that parasitize grapes are not completely clear, and some reported results may need to be further discussed. In Yunnan, we found a high density of RKN-infected grapes, and the pathogenic species was different from previously described RKNs. The morphological

characteristics, such as the perineal pattern, of this species were very similar to those of *Meloidogyne mali*. Therefore, morphological, biochemical and molecular biological methods were used to identify this unknown species.

## Materials and methods

### Nematode population

Samples of grape roots and rhizosphere soils were collected from vineyards in Luliang County, Yunnan Province. Female and egg samples were extracted from the root tissues of grapes. Second-stage juveniles (J2s) were collected from hatching eggs. A portion of the J2s were inoculated on cucumber roots, and the other portion were heat-killed and fixed using a 4% solution of formaldehyde for morphological observation and measurement. Male samples were obtained from the cucumber roots. Some of the males were used for scanning electron microscopy (SEM), and others were heat-killed and fixed using a 4% solution of formaldehyde for morphological observation and measurement. The J2 samples of *M. mali* (used for comparison) were provided by Peng Huan, Institute of Plant Protection, Chinese Academy of Agricultural Sciences.

### Making perineal patterns

Perineal patterns of female adults were made following the method of Xie Hui (2000) [25]. Specifically, female adults were selected from grape root-knot tissue under an anatomical microscope, and a hard plastic consisting of 45% lactic acid solution was used to make an impression of the perineal cuticular pattern with a scalpel. Then, the perineal pattern was cleaned with a 45% lactic acid solution, placed on a glass slide and covered with coverslip, using pure glycerine as a floating carrier.

### Light microscopy (LM)

All nematode samples were observed and examined under a Carl Zeiss Axio Vert. A1 inverted microscope. All samples were measured using the de Man indices [26], and the measurements were expressed in micrometers.

### Scanning electron microscopy (SEM)

The samples of female adults, males and J2s were prepared following the methods of Eisenback *et al*. (1980) [27] and Eisenback and Hirschmann (1979) [28]. The perineal pattern was made following the method of Eisenback *et al*. (1980) [27], with slight modification. Double fixation with 3.5% glutaraldehyde and 1% osmic acid was employed. Specifically, live samples of female adults, males and J2s were cleaned with ddH2O and fixed with a 3.5% glutaraldehyde solution for more than 48 h in a 4˚C refrigerator. After that, they were washed with phosphate-buffered saline (PBS) 3 times, fixed with 1% osmic acid for 2 h, washed with PBS 3 times, dehydrated in a graded ethanol series (30%-100%), critical-point dried, and coated with gold. Observation was performed under a Hitachi S-3000N scanning electron microscope (Japan).

### Isozyme phenotype electrophoresis

Isozyme electrophoresis was carried out following the methods of Esbenshade and Triantaphyllou (1985) [29] and Esbenshade and Triantaphyllou (1990) [30]. Phenotypes were observed for esterases (Est) and malate dehydrogenase (Mdh). Five young egg-laying females of *M. vitis* sp. nov. and five young egg-laying females from a previously identified population of *M. javanica* (used for comparison) were prepared and placed in microtubes containing

**Table 1. The primers used in the research.**

| Primers code | Primer sequence (5′-3′) | References |
|---|---|---|
| 18S | TTGATTACGTCCCTGCCCTTT | Vrain *et al.* (1992) |
| 26S | TCCTCCGCTAAATGATATG | |
| D2A | ACAAGTACCGTGAGGGAAAGTTG | De Ley *et al.* (1999) |
| D3B | TCGGAAGGAACCAGCTACTA | |
| cox1F | TGGTCATCCTGAAGTTTATG | Trinh *et al.* (2019) |
| cox1R | CTACA ACATAATAAGTATCATG | |
| C2F3 | GGTCAATGTTCAGAAATTTGTGG | Powers *et al.* (1993) |
| 1108 | TACCTTTGACCAATCACGCT | |

10 μL mixed liquor consisting of 20% sucrose, 2% Triton X-100 and 0.02% bromophenol blue, the nematodes were broken with a sterile dissecting needle and the enzyme solution can be used immediately or stored at -20˚C refrigerator until use. Electrophoresis was carried out in separating and stacking gels consisted of 7% and 3% polyacrylamide, respectively, 0.75 mm thick, with Tris-glycine buffer (PH8.7) in a Mini-PROTEAN® Tetra Cell apparatus (Bio-Rad). Voltage was maintained at 80 volts for the first 30 minutes, the following was maintained at 150 volts of the separation period until the bromophenol blue dye had migrated to approximately 0.5 cm ahead of the bottom of the gel. Gels was stained with Mdh stain solution for Mdh and with Est stain solution for Est, the preparation of Mdh and Est stain solution following the method of Esbenshade and Triantaphyllou (1985) [31].

## DNA extraction, PCR amplification and sequencing

DNA was extracted from a single female adult and a large number of J2s following the method described by Adam *et al.* (2007) [32] and stored in a -80˚C refrigerator until use. Two rDNA fragments (ITS1-5.8S-ITS2 and D2D3) and two mtDNA fragments (partial coxI and coxII 16S rRNA) of *M. vitis* sp. nov. and *M. mali* were amplified. The primer pairs 18S/26S [33], D2A/D3B [34], cox1F/cox1R [5] and C2F3/1108 [35] were used to amplify the ITS1-5.8S-ITS2 and D2D3 fragments of rDNA and the coxI and coxII fragments of mtDNA, respectively. The primer sequences are listed in Table 1. All of the polymerase chain reactions (PCRs) were performed in 25.00 μL mixed solution containing template DNA (2.50 μL), 10× PCR buffer ($Mg^{2+}$, plus, 2.50 μL), dNTPs (mixture, 2.00 μL), forward and reverse primers (10 μmol/L, 1.00 μL respectively), Taq DNA polymerase (5 U/μL, 0.25 μL), and ddH2O (15.75 μL). All the reagents used in the PCRs were purchased from TransGen Biotech Company. The PCR amplification procedure is provided in Table 2. After the amplification reaction, 5.00 μL PCR product was mixed with 1.00 μL 6× loading buffer (purchased from TaKaRa Company) and electrophoresed in a 1% Tris-acetate-ethylenediaminetetraacetic acid (TAE)-buffered agarose gel. PCR products were excised from the gel and purified using the EasyPure Quick Gel Extraction Kit (purchased from TransGen Biotech Company). The recovered product was ligated with pmD18 cloning

**Table 2. The PCR amplification procedure of primers for the research.**

| Primers | Pre degeneration | Response parameter (35 cycle) | | | Final extension |
|---|---|---|---|---|---|
| | | Degeneration | Annealing | Extension | |
| 18S/26S | 94˚C 4 min | 94˚C 30 s | 55˚C 45 s | 72˚C 1 min | 72˚C 10 min |
| D2A/D3B | 94˚C 4 min | 94˚C 30 s | 60˚C 40 s | 72˚C 1 min | 72˚C 10 min |
| cox1F/cox1R | 94˚C 4 min | 94˚C 30 s | 54˚C 30 s | 72˚C 1 min | 72˚C 10 min |
| C2F3/1108 | 94˚C 4 min | 94˚C 30 s | 51˚C 30 s | 72˚C 1 min | 72˚C 10 min |

vector (purchased from TaKaRa Company) and transformed into DH5α competent cells (purchased from TaKaRa Company). The positive clones were selected and sequenced.

## Phylogenetic analyses and sequence alignment

The obtained sequences were compared with those from other nematodes available in the GenBank database using the BLAST homology search program. ITS1-5.8S-ITS2, D2D3 of rDNA, and partial coxI and coxII 16S rRNA of mtDNA sequences from *Meloidogyne* spp. were selected for phylogenetic reconstruction. The ITS1-5.8S-ITS2 (JX015432.1) sequence of *Rotylenchus buxophilus* and the D2D3 (AY589364.1) sequence of *Ditylenchus halictus* and the coxI (EF617356.1) sequence of *Romanomermis wuchangensis* and complete mitochondrial genome (FN313571.1) sequence of *Radopholus similis* were used as outgroup taxa. A phylogenetic tree was generated based on the neighbor-joining (NJ) method in MEGA 5.1 to analyze the phylogenetic relationships and genetic distances of nematodes. The phylograms were bootstrapped 1000 times to assess the degree of support for phylogenetic analysis.

DNAMAN software was used to compare the ITS1-5.8S-ITS2, D2D3 of rDNA, and partial coxI and coxII 16S rRNA of mtDNA sequences between *M. vitis* sp. nov. and *M. mali*, and analysis the sequence divergence.

## Designing SCAR primers

The rDNA ITS1-5.8S-ITS2 sequences of *M. vitis* sp. nov. amplified in this research were submitted to the GenBank database for BLAST alignment, and a specific sequence in this region was selected to design specific primers in Primer 5.0 software. The primers Mv-F (5-CTGGT TCAGGGTCATTTATAAAC-3) and Mv-R (5-TATACGCTTGTGTGGATGAC-3) were used for PCR amplification of *M. vitis* sp. nov. The PCRs were performed in a 25.00 μL mixed solution containing template DNA for *M. vitis* sp. nov. (2.50 μL), 10× PCR buffer (Mg$^{2+}$, plus, 2.50 μL), dNTPs (mixture, 2.00 μL), forward and reverse primers (10 μmol/L, 1.00 μL respectively), Taq DNA polymerase (5 U/μL, 0.25 μL), and ddH2O (15.75 μL). The amplification procedure was as follows: 4 min at 94˚C, 35 cycles of 30 s at 94˚C, 30 s at 53˚C and 30 s at 72˚C, and a final incubation of 10 min at 72˚C. Amplification products were separated by electrophoresis in a 1% TAE-buffered agarose gel and visualized under ultraviolet light.

## Nomenclatural acts

The electronic edition of this article conformed to the requirements of the amended International Code of Zoological Nomenclature, and hence the new names contained herein are available under that Code from the electronic edition of this article. This published work and the nomenclatural acts it contains have been registered in ZooBank, the online registration system for the ICZN. The ZooBank LSIDs (Life Science Identifiers) can be resolved and the associated information viewed through any standard web browser by appending the LSID to the prefix "http://zoobank.org/". The LSID for this publication is: urn:lsid:zoobank.org:pub: CFA0651C-DDD9-4DA8-A5B2-AF24C7ED8699. The electronic edition of this work was published in a journal with an ISSN and has been archived and is available from the following digital repositories: PubMed Central, LOCKSS.

## Results

*Meloidogyne vitis* sp. nov. Yang, Hu, Liu, Chen, Peng, Wang & Zhang sp. nov. urn: lsid: zoobank. org: act: 0163D840-A867-4F03-9C9C-1F879452E34B.

## Disease symptoms

More than 90% of the grape roots collected from vineyards in Luliang County, Yunnan Province, were seriously damaged by RKNs. The symptoms of lightly nematode-infected plants were not obvious. However, the severely nematode-infected plants presented symptoms of plant dwarfing, leaf yellowing and shedding, little fruit, declining and low growth. The roots were atrophied and distorted, with severe root knots and other symptoms. Both the axial roots and branch roots were damaged. The surface of the infected roots presented numerous galls with white or milky white eggs. Eggs occurred either on the outside of the root galls or within the root galls. The aged roots were rotten and had become necrotic. Adult female heads were found associated with the xylem (Fig 1A and 1B).

## Description of *Meloidogyne vitis* sp. nov.

**Female (n = 25).** The morphometric measurements are shown in Table 3.

*Holotype (female in glycerin).* Body length = 822.99 μm, maximum body width = 531.80 μm, stylet length = 15.25 μm, stylet knob width = 4.29 μm, stylet knob height = 1.85 μm, distance from base of stylet to dorsal esophageal gland opening (DEGO) = 5.32 μm, metacorpus length = 45.50 μm, metacorpus width = 39.49 μm, anterior end to center of metacarpus = 77.64 μm, distance from anterior end to excretory pore = 38.82 μm, distance from anterior end to excretory pore/length of stylet (EP/ST) = 2.5.

*Morphological characters.* The body is pear-shaped and milky white, with a prominent and variably sized neck. The neck has blurry annuluses, the abdomen has slight bulges, the posterior part of the body is round, and the anal region has no protuberances (Figs 2K, 2L and 3E). The stylet is developed, with a straight cone and columnar shaft, the stylet knobs are oblate, the metacorpus is round or ovoid and the valve is developed and obvious, the opening of the dorsal esophageal gland orifice is hook-like, an excretory pore is located in the posterior portion of the stylet knobs (Figs 2J and 3F), and the EP/ST ratio is approximately 1.98–2.96 μm. Under SEM, the labial disc is ovoid-squared, slightly raised on the medial lips, and fused with

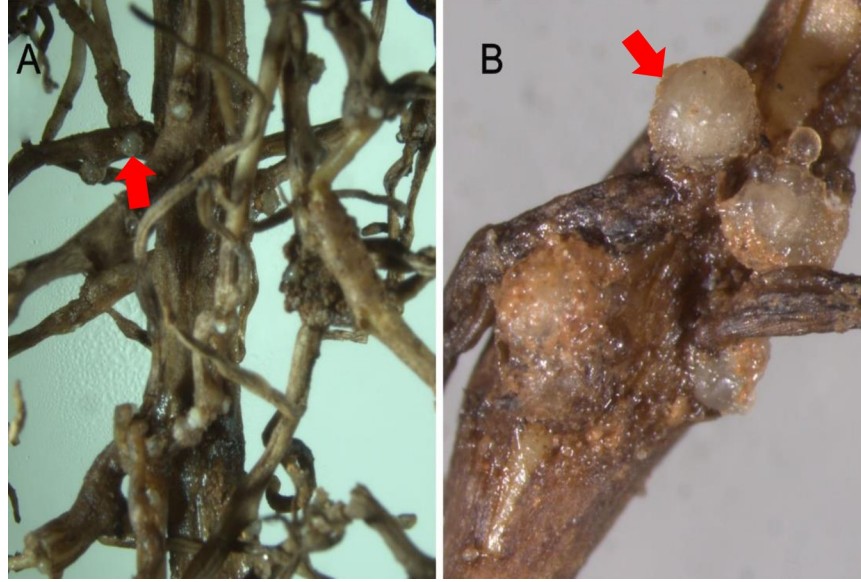

**Fig 1. Root symptom of diseased *Vitis vinifera*.** (A) and (B) all show the root symptom of diseased *Vitis vinifera*, the arrow of fig show eggs of root-knot nematodes.

**Table 3. Morphometrics of *Meloidogyne vitis* sp. nov.**

| Character | Holotype Females | Paratype Females | Males | J2s |
|---|---|---|---|---|
| n | | 25 | 10 | 26 |
| Body length | 822.99 | 958.99 ± 132.32 | 1330.42 ± 179.15 | 396.85 ± 18.34 |
| | | (822.99–1245.16) | (1032.23–1593.38) | (353.36–425.76) |
| Body width | 531.80 | 609.00 ± 43.63 | 36.75 ± 6.15 | 16.19 ± 1.93 |
| | | (531.80–688.11) | (25.69–43.94) | (12.81–22.43) |
| Stylet length | 15.25 | 15.73 ± 3.68 | 19.31 ± 1.71 | 13.33 ± 0.32 |
| | | (8.11–26.58) | (17.02–21.39) | (12.74–14.11) |
| Stylet knobs width | 4.29 | 4.44 ± 0.96 | 3.50 ± 0.62 | 1.56 ± 0.31 |
| | | (2.74–5.95) | (2.65–4.67) | (1.21–2.22) |
| Stylet knobs height | 1.85 | 2.08 ± 0.48 | 2.54 ± 0.29 | 1.24 ± 0.18 |
| | | (1.32–3.32) | (2.23–3.19) | (0.98–1.69) |
| DEGO | 5.32 | 4.13 ± 0.84 | 3.30 ± 0.52 | 1.35 ± 0.31 |
| | | (2.59–5.32) | (2.35–3.91) | (1.02–2.01) |
| Metacorpus length | 45.50 | 42.72 ± 7.05 | 17.95 ± 1.63 | 10.37 ± 1.21 |
| | | (23.01–51.53) | (15.61–20.43) | (8.14–12.18) |
| Metacorpus width | 39.49 | 37.03 ± 5.81 | 9.23 ± 0.73 | 6.94 ± 0.64 |
| | | (21.11–42.86) | (7.92–10.38) | (5.67–8.15) |
| Head region height | / | / | 5.20 ± 0.39 | / |
| | | | (4.70–5.76) | |
| Head region width | / | / | 10.41 ± 1.27 | / |
| | | | (8.33–12.32) | |
| Distance from anterior end to center of metacarpus | 77.64 | 72.75 ± 12.70 | 99.31 ± 5.88 | 54.89 ± 1.99 |
| | | (44.17–86.28) | (90.96–108.73) | (50.8–58.62) |
| Distance from anterior end to excretory pore | 38.82 | 38.82 ± 4.15 | 133.33 ± 4.94 | 41.55 ± 2.13 |
| | | (34.33–44.80) | (126.49–140.81) | (37.46–45.31) |
| Tail length | / | / | 12.86 ± 0.77 | 57.43 ± 3.92 |
| | | | (11.81–14.23) | (47.01–63.77) |
| Hyaline tail length | / | / | / | 12.16 ± 1.74 |
| | | | | (9.72–15.73) |
| Anal body diameter | / | / | 24.05 ± 1.81 | 12.76 ± 1.91 |
| | | | (21.68–26.68) | (10.15–17.11) |
| Spicules length | / | / | 30.88 ± 2.59 | / |
| | | | (27.86–35.75) | |
| Gubermaculum length | / | / | 10.23 ± 1.86 | / |
| | | | (8.15–14.88) | |
| a (Body length/ Body width) | 1.55 | 1.58 ± 0.2 | 36.79 ± 5.96 | 24.76 ± 2.51 |
| | | (1.30–1.95) | (30.67–50.15) | (18.98–28.44) |
| c (Body length/ Tail length) | / | / | 103.59 ± 13.78 | 6.95 ± 0.65 |
| | | | (81.72–127.65) | (6.15–8.77) |
| d (Tail length / Anal body diameter) | / | / | 0.54 ± 0.03 | 4.58 ± 0.60 |
| | | | (0.48–0.57) | (3.39–5.34) |
| Tail length/Hyaline tail length | / | / | / | 4.81 ± 0.76 |
| | | | | (3.44–6.08) |

All measurements are in μm and shown in the form: mean ± s.d. (range).

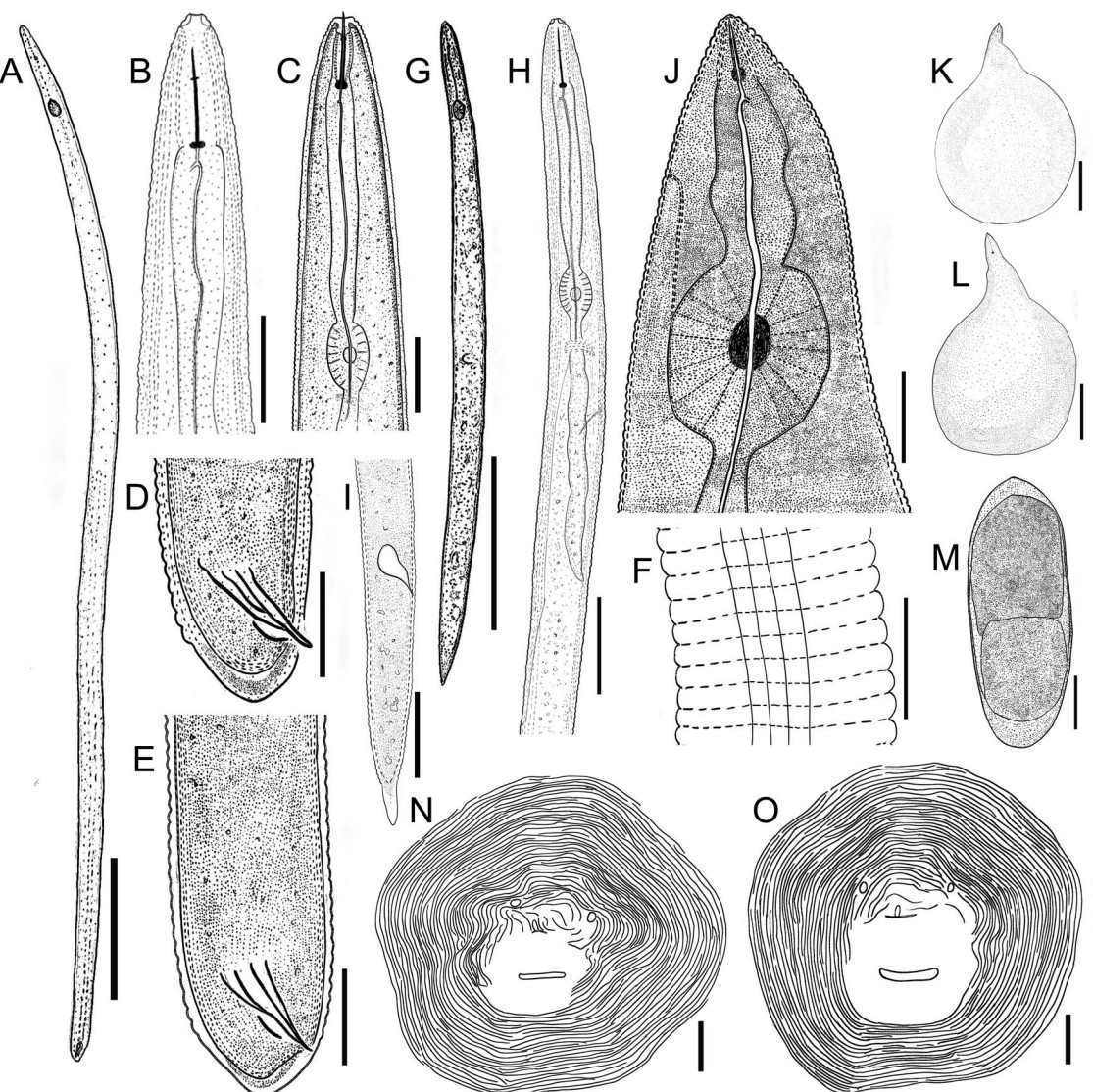

**Fig 2. Line drawings of *Meloidogyne vitis* sp. nov.** Male (A-F) A: Entire body of male; B, C: Anterior region of male; D, E: Tail region of male; F: Lateral field of male. Second-stage juveniles (G-I) G: Entire body of second-stage juveniles; H: Anterior region of second-stage juveniles; I: Tail region of second-stage juveniles. Female (J-O) J: Anterior region of female; K, L: Entire body of female; M: Eggs of female; N, O: Perineal pattern of female. (Scale bars: A, K, L = 200 μm; B-E, I, H, G, M-O = 20 μm; G = 100 μm; F = 10 μm).

the medial lips to form a dumbbell-shaped structure; there are no obvious lateral lips, and the oral aperture is slit-like and located in the middle of the labial disc, surrounded by six inner labial sensilla (Fig 4B). An excretory pore is located on ventrally region between 23rd and 25th annule behind lips (Fig 4A and 4E). The stylet is cone-shaped and sharply pointed (Fig 4D).

The perineal pattern of female adults is round to ovoid with a moderately high dorsal arch and smooth and fine striae that are extremely dense and faint; lateral fields are not clearly visible, and there are no lateral lines, however, a few specimens have slight striae on two shoulders or wings in the lateral field; two phasmids are large, prominent and round, with a diameter that can account for 2–5 annular striae, and seemingly eye-shaped; straight lines of two phasmids are parallel or nearly parallel to the vulva; the vulval slit is wide and seemingly mouth-

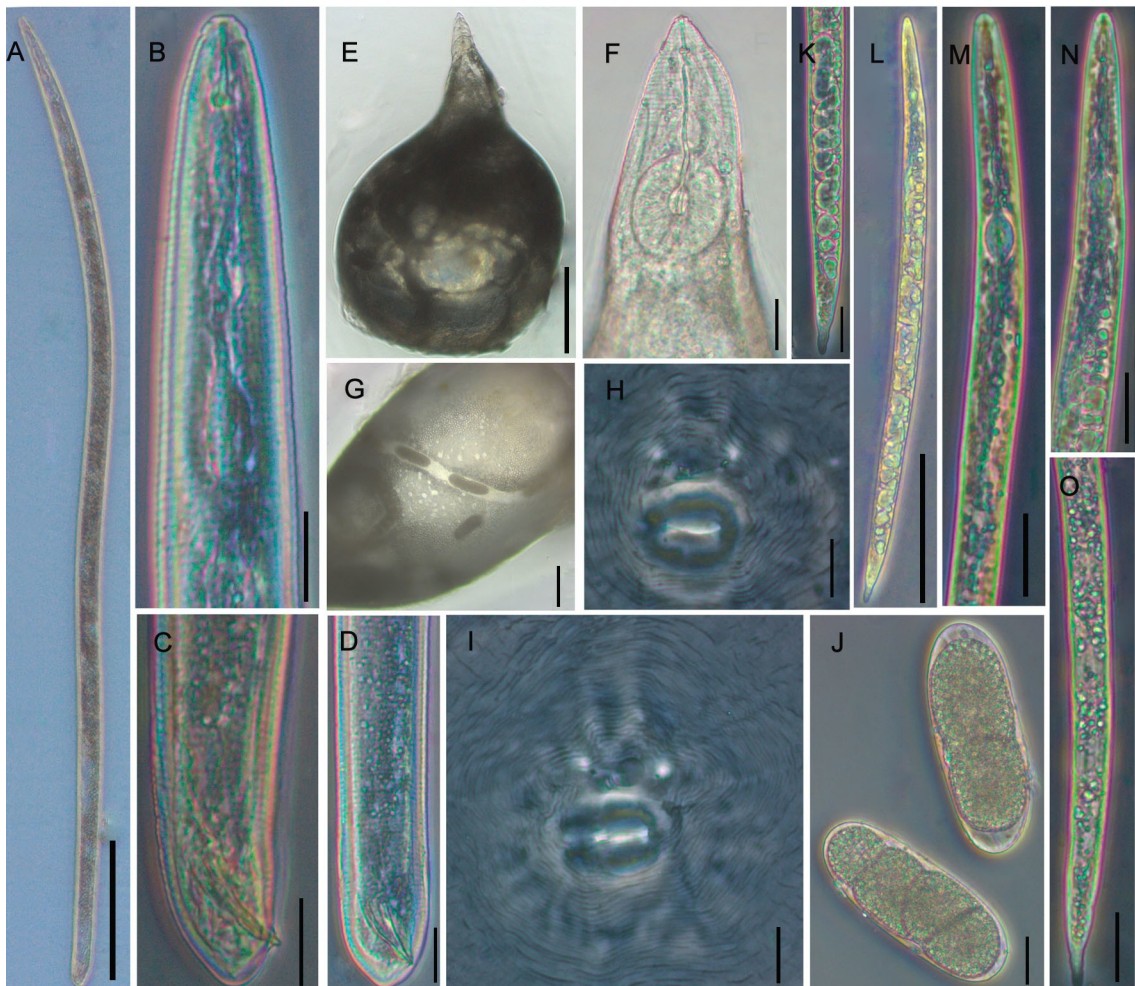

**Fig 3. Light micrographs of *Meloidogyne vitis* sp. nov.** Male (A-D) A: Entire body of male; B: Anterior region of male; C, D: Tail region of male. Female (E-J) E: Entire body of female; F: Anterior region of female; G: Partial region of female; H, I: Perineal pattern of female; J: Eggs of female. Second-stage juveniles (K-O) K, O: Tail region of second-stage juveniles; L: Entire body of second-stage juveniles; M, N: Anterior region of second-stage juveniles (Scale bars: A, E = 200 μm; B-D, F, H, I, J, K, M-O = 20 μm; G, L = 100 μm).

shaped; the anal fold is clearly visible and seemingly nose-shaped; the whole perineal pattern is seemingly monkey-face-shaped; the area of the vulva and anus is smooth, with no striae (Figs 2N, 2O, 3H and 3I). The distance between two phasmids is wider than or equal to the length of the vulval slit; however, in very few specimens, this value is slightly smaller. The vulva-anus distance is short: 19.94±1.63 (17.35–23.48) μm. The vulva-phasmid distance is 27.98 ± 2.33 (23.28–33.04) μm. The anus-phasmid distance is 6.84 ± 1.49 (4.73–9.79) μm. The morphology of the perineal pattern under SEM is consistent with that under LM, but SEM shows more morphological details of the anus and vulva, and the striae are clearer (Fig 4C).

**Male (n = 10).** The morphometric measurements are shown in Table 3.

*Morphological characters*. The body is vermiform and variable in length, and the anterior end of the body tapers off (Figs 2A and 3A). The head cap is obvious and slightly separated from the body; the stylet is developed and has an obvious boundary with the stylet shaft; the stylet knot is oblate-spheroidal; the metacorpus is vertically ovoid; and the valve is obvious (Figs 2C and 3B). The tail is mostly straight and short with a humped end; the spicules are

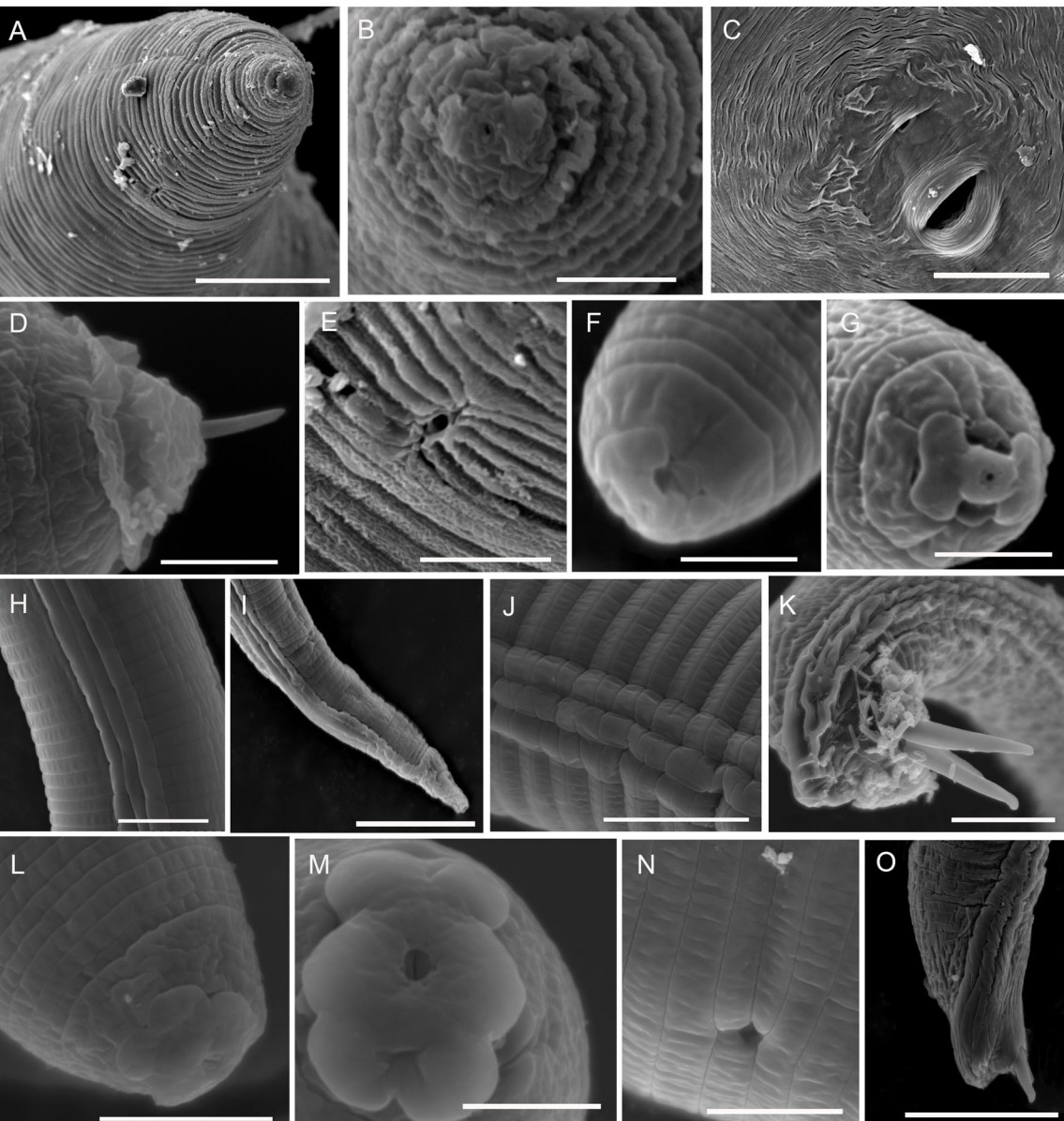

**Fig 4. Scanning electron microscope photographs of *Meloidogyne vitis* sp. nov.** Female (A-E) A: anterior end in lateral view; B: anterior end in face view; C: perineal pattern; D: anterior end in lateral view and see the stylet; E: excretory pore. Second-stage juvenile (F-I) F: anterior end in lateral view; G: anterior end in face view; H: lateral field; I: tail region. Male (J-O) J: lateral field; K: tail region; L: anterior end in lateral view; M: anterior end in face view; N: excretory pore; O: tail region. (Scale bars: A, C, O = 20 μm; B, D, M = 3 μm; F, G = 2 μm; E, L, H, N = 5 μm; J, I, K = 10 μm).

developed, arch-shaped, and slightly curved; and the gubernaculum is obvious and curved-moon shaped (Figs 2D, 2E, 3C and 3D). Under SEM, the head region lacks annulus; the labial disc is horizontally ovoid-squared, slightly raised on the medial lips, slightly wider than the medial lips and fused with the medial lips to form a dumbbell-shaped structure; there are no lateral lips; a prominent slit-like opening between the labial disc and medial lips is observed; a distinct depression appears in the middle of the medial lips; and the oral aperture is slit-like and located in the middle of the labial disc, surrounded by six inner labial sensilla (Fig 4L and

4M). The lateral field consists of four incisures forming 3 lateral bands, which are full of reticular striae (Fig 4J). Lateral incisures extend to the tail end of the body, and in addition to the tail end, the annulus passes through incisures. The spicules resemble a figure eight, and the tip end slightly curves to form a hook-like shape (Fig 4K and 4O). The excretory pore is an irregular pore located in the depression of the cuticle, where it spans two annuluses, and the regular body annuluses are interrupted around the excretory pore (Fig 4N).

**J2 (n = 26).** The morphometric measurements are shown in Table 3.

*Morphological characters*. The body is vermiform and slender and tapers at both ends, but more towards the tail than the anterior end (Figs 2G and 3L). The body annulations are not obvious. The stylet is straight, slender, and sharply pointed and has an obvious boundary with the stylet shaft; the stylet knobs are obvious and spherical; and the metacorpus is ovoid and clearly visible (Figs 2H, 3M and 3N). The tail is variable, exhibits a range of variation in tail fields and is conical and constricted; the hyaline tail is short (Figs 2I, 3K and 3O). The anus is difficult to distinguish except under a high-power oil immersion objective lens. The intestinal contents were too much, so the rectum and caudal sensory organ were difficult to observe. Under SEM, the head region is not smooth and slightly folded; the labial disc appears round, slightly raised on the medial lips and fused with the medial lips to form a dumbbell-shaped structure, with a slightly sunken appearance in the middle of the medial lips; a prominent slit-like amphidial opening is located between the labial disc and lateral lips; the oral aperture is round and located in the middle of the labial disc, surrounded by six inner labial sensilla (Fig 4F and 4G). The lateral field forms 3 lateral bands delimited by four incisures (Fig 4H). The anal opening is elliptical and located in the cuticular depression in the tail of the body (Fig 4I). The excretory pore is irregular and located in the cuticular depression, and the regular body annuluses are interrupted around the excretory pore.

**Egg (n = 20).** Eggs of female adults are oval-shaped (Figs 2M and 3J). Twenty eggs were measured in clear water. The egg length was 86.52–110.69 μm (mean: 98.67 μm, standard error: 6.44), and the egg width was 30.56–39.31 μm (mean: 34.73 μm, standard error: 2.38).

## Taxonomic summary

**Type host.** Grape (*Vitis vinifera* L., *Vitis* L., *Vitaceae*).

**Type locality.** Luliang County, Yunnan Province, China (25˚07' N, 103˚78' E).

**Etymology.** The specific epithet refers to the host plant on which this new species was found. Additionally, because of its ability to seriously infect cultivated grapes (*V. vinifera* L.), we suggest the name *Meloidogyne vitis* sp. nov.

**Type material.** The holotype female and paratypes, males, perineal patterns and J2s were deposited in the nematode collection of the authors' Laboratory of Plant Nematology, College of Plant Protection, Yunnan Agricultural University, China. Specific ITS1-5.8S-ITS2, D2D3 rDNA, coxI rRNA and coxII 16S rRNA mtDNA sequences were deposited in GenBank with accession numbers MN816222.1, MN816223.1, MN816225.1, MN816226.1, MN814829.1, MN814830.1, MT012386.1, and MT012387.1, respectively.

## Diagnosis and relationships

*Meloidogyne vitis* sp. nov. can be distinguished from other *Meloidogyne* spp. by a unique combination of several morphological characters. The perineal pattern of female adults is round or ovoid, with large and prominent phasmids. Females have a prominent neck, an excretory pore is located on ventrally region between 23rd and 25th annule behind lips, and the EP/ST ratio is approximately 2.5 (1.98–2.96 μm). The male has a prominent head region, the labial disc is fused with the medial lips to form a dumbbell-shaped structure, a wide slit is located between

the labial disc and medial lips, the tail is blunt and round, the gubernaculum is prominent and arch-shaped, and the lateral field consists of four incisures. J2s are characterized by their head region is not smooth and slightly folded, the labial disc is fused with the medial lips to form a dumbbell-shaped structure, the hyaline tail is short and constricted, the relatively small c value (6.15–8.77 μm), and the lateral fields have four incisures. In addition, *M. vitis* sp. nov. has unique ITS1-5.8S-ITS2 and D2D3 of rDNA, coxI and coxII 16S rRNA of mtDNA sequences.

Because of the large and prominent phasmids in the perineal pattern in female adults, *M. vitis* sp. nov. is similar to *M. mali* [36], *Meloidogyne artiellia* [37], *Meloidogyne floridensis* [38], *Meloidogyne naasi* [39], *M. nataliei* [18], *Meloidogyne shunchangensis* [40], *Meloidogyne kongi* [41], *Meloidogyne dimocarpus* [42], and *Meloidogyne thailandica* [43]. *Meloidogyne vitis* sp. nov. differs from *M. mali* in that the perineal pattern of the female shows a moderately high dorsal arch rather than a low and flat dorsal arch and there are no lateral lines rather than clear single or double lateral lines in the lateral fields, the DEGO of male and J2 are smaller (2.35–3.91 vs. 6.00–13.00 μm and 1.02–2.01 vs. 4–6 μm, respectively), the J2 tail is longer (47.01–63.77 vs. 30–34 μm), and the J2 c value is smaller (6.15–8.77 vs. 12–15 μm). *Meloidogyne vitis* sp. nov. differs from *M. artiellia* in the lip region of the female (the lateral lip is not obvious rather than appearing as six almost equally sized lips), the greater body length (822.99–1245.16 vs. 650–760 μm) and body width (531.80–688.11 vs. 340–460 μm) in females, the perineal pattern of the female being round or ovoid rather than the general outline pattern resembling a figure eight, the different male lip region (the labial disc and medial lips are fused into a dumbbell-shaped structure instead of the lip region appearing as six nearly equally sized lips), its smaller male DEGO (2.35–3.91 vs. 5.00–7.00 μm), its longer J2 tail (47.01–63.77 vs. 24.5 μm), its smaller J2 stylet length and c value (12.74–14.11 vs. 14–16 μm and 6.15–8.77 vs. 13–16 μm, respectively). *Meloidogyne vitis* sp. nov. differs from *M. floridensis* in that its perineal pattern of the female has no lateral lines rather than faint lateral lines, it has narrower stylet knobs in males (2.65–4.67 vs. 5.00–6.00 μm), and it has a smaller J2 DEGO (1.02–2.01 vs. 2.50–3.00 μm). *Meloidogyne vitis* sp. nov. differs from *M. nassi* in that the posterior of the female is smooth rather than presenting a slight protuberance, the excretory pore of the female is situated behind instead of slightly in front of the stylet knobs, the longer females body length (822.99–1245.16 vs. 455–705 μm) and body width (531.80–688.11 vs. 227–398 μm), and it lacks the four or five small and vesicle-like structures grouped irregularly round in front of the metacorpus that can be found in *M. nassi* male and J2. *Meloidogyne vitis* sp. nov. differs from *M. nataliei* in that the posterior of the female is smooth rather than having a slight protuberance; the perineal pattern of the female has no lateral lines instead of two separated ropelike striae extending laterally from the vulval and anal areas and forming a lateral field; the stylet length, DEGO, and spicule length of male are smaller (17.02–21.39 vs. 28.40–29.20 μm, 2.35–3.91 vs. 4.0–6.5 μm, and 27.86–35.75 vs. 41.3–44.3 μm respectively); and the body length, stylet length, and DEGO of J2 are smaller (353.36–425.76 vs. 539.00–641.00 μm, 12.74–14.11 vs. 21.9–22.8 μm, and 1.02–2.01 vs. 3.0–4.3 μm, respectively). *Meloidogyne vitis* sp. nov. differs from *M. shunchangensis* in the female lip region (the lateral lip is not obvious rather than the lip region appearing as six lips), the perineal pattern of females having no striae rather than occasionally having lines of striae between the anus and vulva and having no incisures rather than sometimes having obvious double striae in the lateral field, the greater J2 tail length and transparent tail length (47.01–63.77 vs. 20.3–28.6 μm and 9.72–15.73 vs. 3.4–4.2 μm, respectively), the smaller J2 DEGO and c value (1.02–2.01 vs. 2.6–3.7 μm and 6.15–8.77 vs. 11.9–16.60 μm, respectively). *Meloidogyne vitis* sp. nov. differs from *M. kongi* in that the female's posterior is smooth rather than possessing a prominent protuberance, the perineal pattern is no line striae vs full of line striae between the anus and vulva, the female body is longer (822.99–1245.16 vs. 610.6–820.3 μm), the DEGO of J2 is smaller (1.02–2.01 vs. 3.9–5.8 μm),

there are no lateral lips rather than two semicircular lateral lips in the head region in males, the stylet length and DEGO of male are smaller (17.02–21.39 vs. 22.00–23.90 μm and 2.35–3.91 vs. 5.80–7.50 μm, respectively), and the DEGO of J2 is smaller (1.02–2.01 vs. 3.9–5.8 μm). *Meloidogyne vitis* sp. nov. differs from *M. dimocarpus* in that the perineal pattern of the female has no lateral lines instead of double or single striae in the lateral field and no striae instead of multiple continuous striae between the anus and vulva, the phasmids are large instead of small, and the male and J2 DEGO are smaller (2.35–3.91 vs. 4.50–5.75 μm and 1.02–2.01 vs. 2.25–3.75 μm, respectively). *Meloidogyne vitis* sp. nov. differs from *M. thailandica* in that the perineal pattern of the female lacks the radial structures underneath the pattern area that are characteristic of *M. thailandica*, the J2 DEGO and hyaline tail length are smaller (1.02–2.01 vs. 2.5–3.5 μm and 9.72–15.73 vs. 15–20 μm, respectively).

*Meloidogyne vitis* sp. nov. can also be distinguished from several other *Meloidogyne* species infecting grape, including *M. incognita*, *M. javanica*, *M. arenaria*, *M. hapla*, *M. ethiopica* and *M. thamesi*, by the large and prominent phasmids in the perineal pattern of females.

## Isozyme analysis

The isozyme electrophoretic analysis of young egg-laying females of *M. vitis* sp. nov. showed three rare Mdh bands (Fig 5A, MV lane) and one rare Est band migrating rapidly in the gel (Fig 5B, MV lane), which did not occur in the Mdh and Est phenotypes of *M. javanica*. The Mdh and Est bands of *M. vitis* sp. nov. have not been reported in other RKNs. According to the relative mobility (Rm) values and referring to the naming method of Esbenshade and Triantaphyllou (1985) [29], the Mdh band was named N3d, and the Est band was named VF1.

## PCR product electrophoresis

Size of the PCR amplification bands in different fragments of *M. vitis* sp. nov. and *M. mali* as follows: for both of the ITS1-5.8S-ITS2 fragments was approximately 870 bp, both of the 28S D2D3 fragments was approximately 770 bp, both of the coxI mtDNA fragments was approximately 400 bp; the coxII mtDNA fragments of *M. vitis* sp. nov. was approximately 550 bp (Fig 6).

## Molecular characterization

Amplification and sequencing of the ITS1-5.8S-ITS2 fragment of rDNA from the females and J2s of *M. vitis* sp. nov. and J2s of *M. mali* revealed that the sequence sizes were 877 bp, 877 bp, and 873 bp, respectively. The GenBank accession numbers are MN816222.1 and MN816223.1 for the female and J2 of *M. vitis* sp. nov., respectively, and MN816224.1 for the J2 of *M. mali*. The ITS1-5.8S-ITS2 fragment identities were 100% for the female and J2 of *M. vitis* sp. nov. A BLAST search of *M. vitis* sp. nov. revealed that the most similar sequence was that of *M. mali* (GenBank accession numbers KR535971.1, JX978229.1, and JX978228.1), with an identity of only 87%. The sequence of *M. mali* identified in this research was most similar to sequences of *M. mali* in GenBank, with identities ranging from 97% (accession number KR535971.1) to 99% (accession numbers JX978225.1, JX978229.1, and JX978228.1). Phylogenetic trees (52 sequences in total) showed that the female and J2 of *M. vitis* sp. nov. formed a well-supported clade with high bootstrap support (100%) and were closest to *M. mali* from GenBank (accession numbers KR535971.1, JX978229.1, and JX978228.1) and *M. mali* identified in this research (accession number MN816224.1), all of which formed one group with high bootstrap support (95%). *Meloidogyne vitis* sp. nov. was clearly separated from other species (Fig 7). Sequence alignment of ITS1-5.8S-ITS2 rDNA between female of *M. vitis* sp. nov. and J2 of *M. mali* identified in this research showed that the identity was only 77.09%, and the sequences were thus highly diverged (210-base divergence) (S1 Fig).

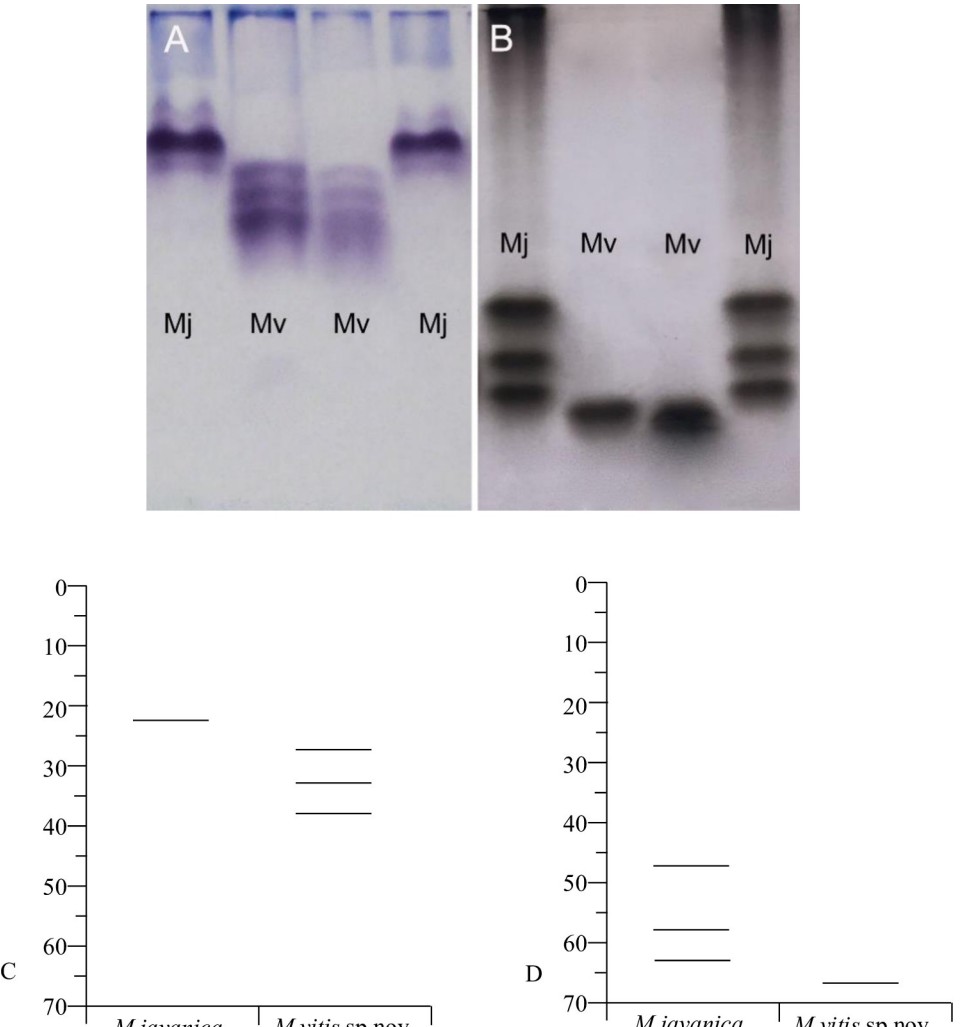

**Fig 5. Malate dehydrogenase and esterase phenotype patterns obtained with electrophoresis of protein homogenates from five young egg-laying females of *Meloidogyne vitis* sp. nov. (lane Mv) and five young egg-laying females of the *Meloidogyne javanica* reference population (lane Mj).** (A) Malate dehydrogenase patterns. (B) esterase patterns. (C) Relative mobility (Rm) of malate dehydrogenase bands. (D) Relative mobility (Rm) of esterase bands.

Amplification and sequencing of the D2D3 fragment of 28S rDNA from females and J2s of *M. vitis* sp. nov. revealed sequence sizes both were 775 bp. The GenBank accession numbers are MN816225.1 and MN816226.1 for the female and J2 of *M. vitis* sp. nov., respectively. The D2D3 fragment identities were 99.61% for the female and J2 of this new species. A BLAST search of *M. vitis* sp. nov. revealed that the most similar sequence was that of *M. mali* (GenBank accession numbers KX430177.1, JX978226.1, JX978227.1, and KF880398.1), with a 93% identity. Phylogenetic trees (45 sequences in total) showed that the female and J2 of *M. vitis* sp. nov. formed a well-supported clade with high bootstrap support (99%). *Meloidogyne vitis* sp. nov. is most closely related to *M. mali* from GenBank (accession numbers KF880398.1, JX978227.1, KF880399.1, KX430177.1, and JX978226.1) and formed a sister group with this species with high bootstrap support (98%). *Meloidogyne vitis* sp. nov. was clearly separated from other species (Fig 8). Squence alignment of D2D3 28S rDNA between female of *M. vitis*

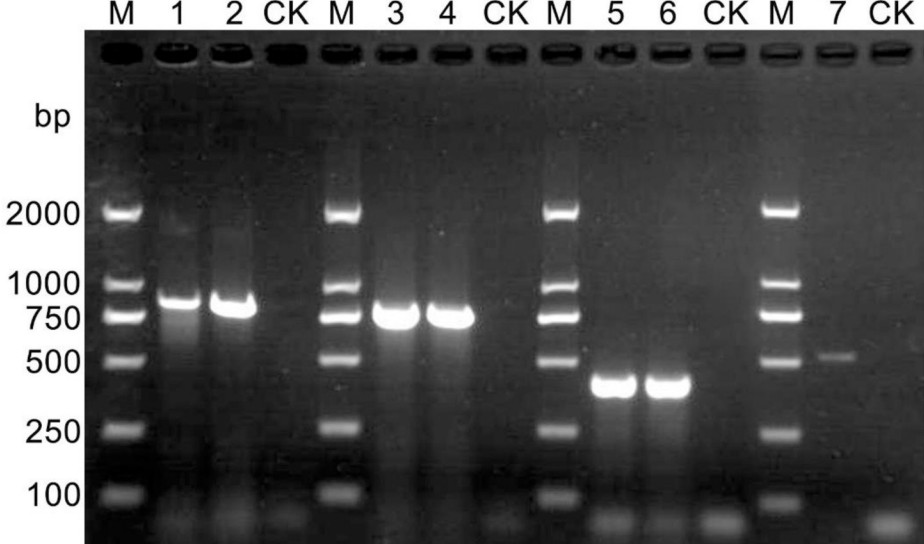

**Fig 6. PCR electropherogram for different fragments of *Meloidogyne vitis* sp. nov. and *Meloidogyne mali*.** M: 2000 DNA marker; CK: The negative control consisting of water; Lanes 1–2: The ITS1-5.8S-ITS2 region of *Meloidogyne vitis* sp. nov. and *Meloidogyne mali*, respectively; Lanes 3–4: The D2/D3 region of *Meloidogyne vitis* sp. nov. and *Meloidogyne mali*, respectively; Lanes 5–6: The coxI region of *Meloidogyne vitis* sp. nov. and *Meloidogyne mali*, respectively; Lane 7: The coxII region of *Meloidogyne vitis* sp. nov.

sp. nov. and *M. mali* from GenBank (accession number KX430177.1) showed that the identity was 93.81% and the sequences were highly divergent (48-base divergence) (S2 Fig).

The sequence sizes of the coxI fragments of 16S rRNA from the female and J2s of *M. vitis* sp. nov. and J2s of *M. mali* were 413 bp, 413 bp, and 417 bp, respectively. The GenBank accession numbers are MN814829.1 and MN814830.1 for the female and J2 of *M. vitis* sp. nov., respectively, and MN814831.1 for the J2 of *M. mali*. The sequence identities were 99.76% for the female and J2 of *M. vitis* sp. nov. A BLAST search of *M. vitis* sp. nov. revealed the highest match with the sequences of *Meloidogyne ichinohei* (GenBank accession number KY433448.1) and *M. exigua* (GenBank accession numbers MH128478.1, MH128477.1, and MH128476.1), with identities of 85%. The sequences of *M. mali* identified in this research were most similar to those of *M. mali* from GenBank (accession numbers KM887146.1, KM887145.1, KY433450.1 and KY433449.1), with 99% identities. Phylogenetic trees (34 sequences in total) showed that the female and J2 of *M. vitis* sp. nov. formed a well-supported clade with high bootstrap support (99%). *Meloidogyne vitis* sp. nov. was most closely related to *M. mali* from GenBank (accession numbers KM887146.1, KY433450.1, KM887145.1, and KY433449.1) and *M. mali* identified in this research (accession number MN814831.1), and they formed a sister group. *Meloidogyne vitis* sp. nov. was clearly separated from other species (Fig 9). Sequence alignment of coxI 16S rRNA between the female of *M. vitis* sp. nov. and J2 of *M. mali* identified in this research showed that the identity was 84.26% and the sequences were highly divergent (67-base divergence) (S3 Fig).

The coxII 16S rRNA sequences from the female and J2s of *M. vitis* sp. nov. were 545 bp and 540 bp in size, respectively. The GenBank accession numbers are MT012386.1 for the female and MT012387.1 for the J2 of *M. vitis* sp. nov. The identities were 99.63% for the female and J2 of *M. vitis* sp. nov. A BLAST search revealed that *M. vitis* sp. nov. is most closely related to *M. mali* (GenBank accession number KC112913.1), with an identity of 81%. Phylogenetic trees (53 sequences in total) showed that the female and J2 of *M. vitis* sp. nov. formed a well-supported clade with high bootstrap support (100%). *Meloidogyne vitis* sp. nov. was most closely related to *M. mali* from GenBank (accession number KC112913.1) and formed one

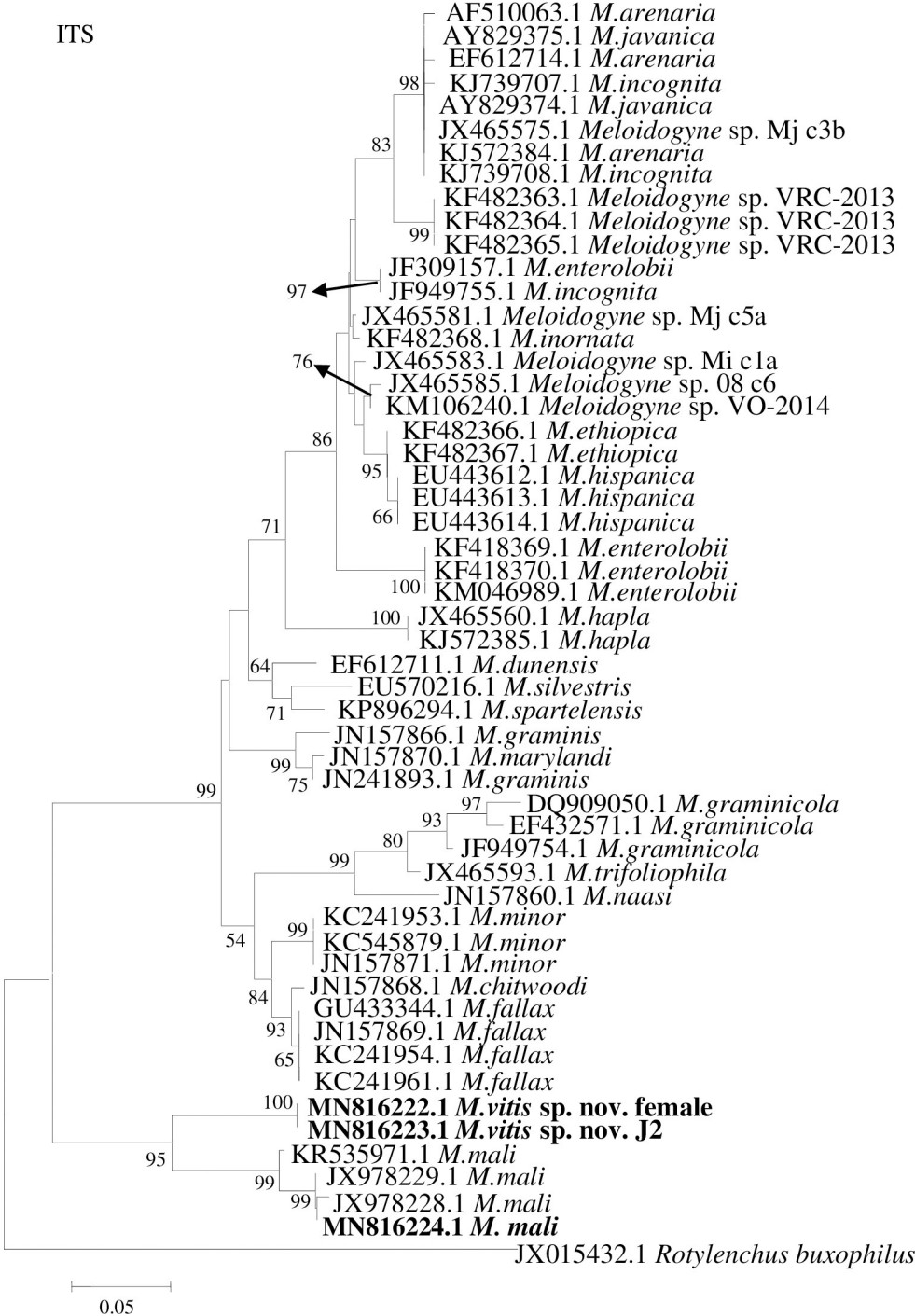

**Fig 7. Phylogenetic relationships of *Meloidogyne vitis* sp. nov. with other root-knot nematodes based on ITS1-5.8S-ITS2 sequences.** Numbers to the left of the branches are bootstrap values for 1000 replications.

monophyletic clade with moderate bootstrap support (74%) (Fig 10). Sequence alignment of coxII 16S rRNA between female of *M. vitis* sp. nov. and *M. mali* from GenBank (accession number KC112913.1) showed that the identity was only 78.58% and the sequences were highly divergent (118-base divergence) (S4 Fig).

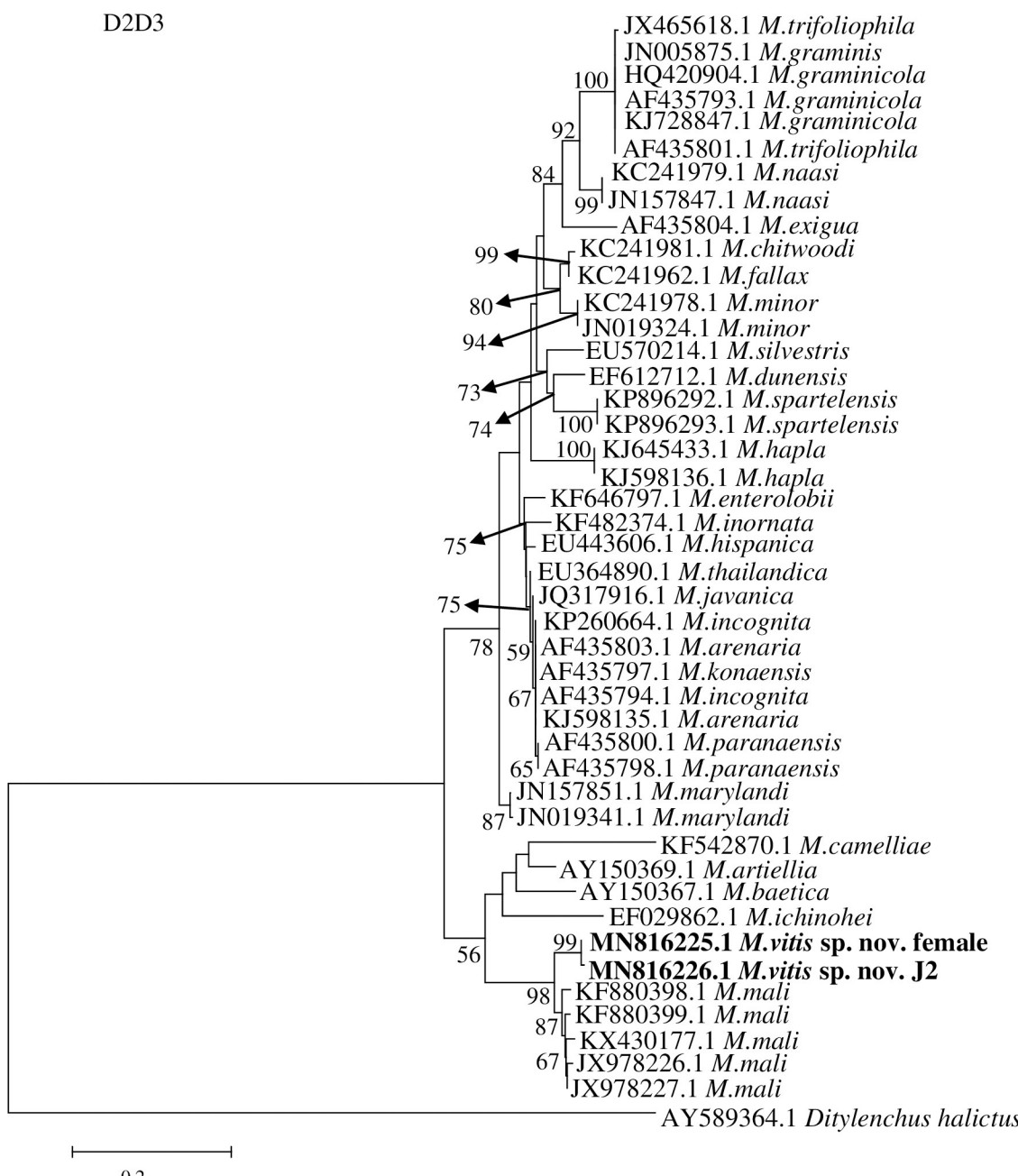

**Fig 8. Phylogenetic relationships of *Meloidogyne vitis* sp. nov. with other root-knot nematodes based on D2/D3 sequences of 28S rDNA.** Numbers to the left of the branches are bootstrap values for 1000 replications.

## SCAR-PCR analysis

Individual females of previously identified and purified populations of *M. incognita*, *M. javanica*, *M. arenaria*, *M. hapla*, and *Meloidogyne enterolobii* were used for comparison. DNA was extracted from individual females of *M. incognita*, *M. javanica*, *M. arenaria*, *M. hapla*, *M. enterolobii* and *M. vitis* sp. nov. The ITS1-5.8S-ITS2 fragment of the six RKN species was amplified using the primers 18S/26S. In all populations, a single band with a size of approximately 760–900 bp was amplified (Fig 11A). However, using the primers Mv-F/Mv-R to

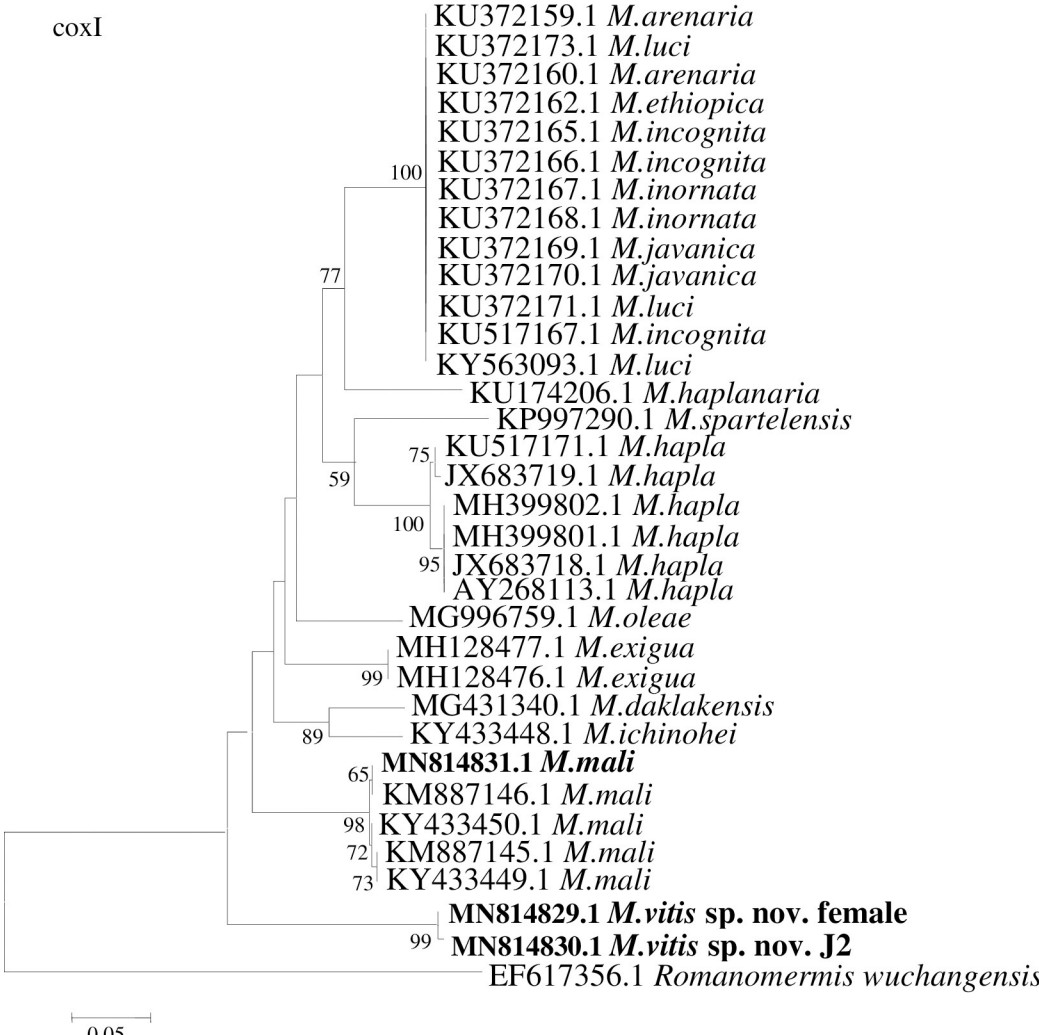

coxI

**Fig 9. Phylogenetic relationships of *Meloidogyne vitis* sp. nov. with other root-knot nematodes based on coxI-rRNA genes sequences.** Numbers to the left of the branches are bootstrap values for 1000 replications.

amplify the same six templates mentioned above, species-specific fragments of approximately 170 bp were amplified only in *M. vitis* sp. nov., no fragments were observed for templates from the other five RKN species (Fig 11B). The species-specific product of *M. vitis* sp. nov. was recycled, cloned and sequenced, resulting in a 174 bp sequence, which was deposited in the GenBank database for BLAST alignment, no similar sequences were found. All results indicated that the Mv-F/Mv-R primers were specific and reliable.

## Discussion

Reliable detection and identification technology is necessary for the protection of agricultural production systems against quarantine nematodes worldwide [44]. In the past, RKNs were identified mainly by morphological observations. Although observations of the perineal pattern of female adults is the primary method used for morphological identification, there is some intraspecific variability in this pattern due to differences in host and nutrition, differences between young females and female adults and other factors. Therefore, identification results are often uncertain.

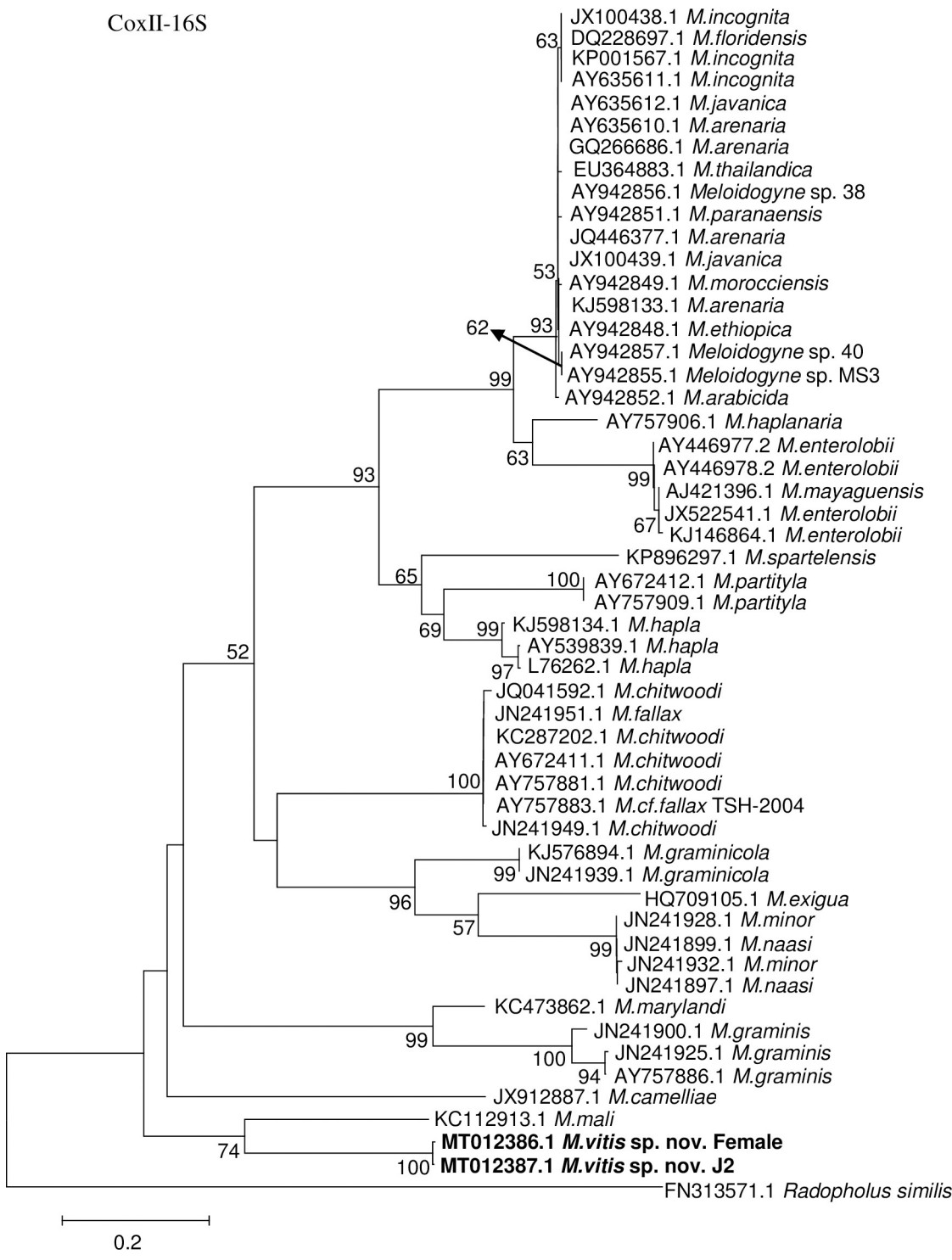

CoxII-16S

**Fig 10. Phylogenetic relationships of *Meloidogyne vitis* sp. nov. with other root-knot nematodes based on coxII-16S rRNA genes sequences.** Numbers to the left of the branches are bootstrap values for 1000 replications.

The perineal pattern of *Meloidogyne inornata* is similar to that of *M. incognita*, making it difficult to distinguish these two species [45], and the perineal pattern of pre-adult *M. javanica* resembles that of *Meloidogyne africana* adults [46]. In addition, the perineal pattern of RKNs varies greatly

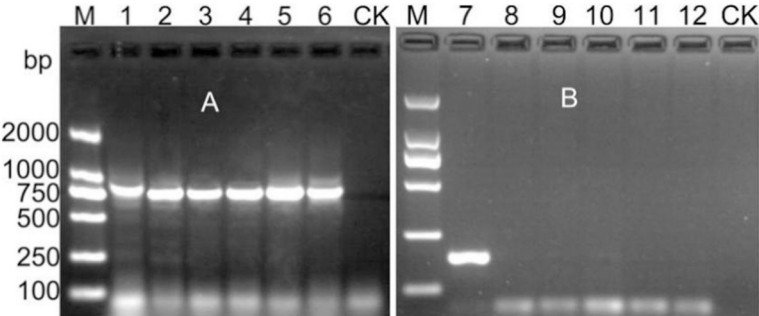

**Fig 11. PCR amplification of the supplied RKNs with the Mv-F/Mv-R primers.** M: 2000 DNA marker; CK: The negative control consisting of water. A: Lanes 1–6, The ITS1-5.8S-ITS2 region of *Meloidogyne vitis* sp. nov., *Meloidogyne incognita*, *Meloidogyne javanica*, *Meloidogyne arenaria*, *Meloidogyne hapla* and *Meloidogyne enterolobii*, respectively. B: Lanes 7–12, The amplification results of root-knot nematode species-specific PCRs of *Meloidogyne vitis* sp. nov., *Meloidogyne incognita*, *Meloidogyne javanica*, *Meloidogyne arenaria*, *Meloidogyne hapla* and *Meloidogyne enterolobii*, respectively.

across generations; for example, *M. javanica* has a total variation rate of 22.6%; the variation is mainly caused by nonobvious incisures and the formation of shoulder protuberances, which make it difficult to distinguish this RKN from *M. arenaria* [47]. Traditional morphological methods face considerable challenges in the identification of RKNs due to intraspecific variation and interspecies similarity [48]. Therefore, PCR-based methods and biochemical methods are becoming increasingly important in the diagnosis of *Meloidogyne* spp.

The technology of isozyme (in particular Est and Mdh) electrophoretic is a relatively old method for the identification of *Meloidogyne* spp., but it is still used by many researchers to identify some RKNs. This technique remains as an effective methodology with which to unambiguously identify and differentiate *Meloidogyne megadora* [49]. In this research, the band phenotypes of malate dehydrogenase from the new species were different from those of other RKNs reported to date; the new species produced three bands and showed the N3d phenotype. The esterase of the new species migrated rapidly in the gel, showing a VF1 phenotype, which is the same esterase phenotype observed for *M. naasi*, *Meloidogyne exigua* and *M. thailandica*; however, all of these species have different malate dehydrogenase phenotypes [39, 50]. Nevertheless, the isozyme analyses applied in species identification are limited because they are effective only when egg-laying females are available [51].

PCR-based methodologies are of ever-increasing importance in species diagnostics and phylogenetics within the genus *Meloidogyne* [52]. Phylogenetic analyses of rDNA sequences are considered a reliable diagnostic approach and are commonly used to identify and compare certain RKNs [53]. Based on rDNA-PCR identification, the ITS fragment is possibly the most widely used genetic marker for living organisms and the most commonly used species-level marker used for organisms (plants, protists, and fungi) [52]. This fragment has been widely used to identify RKNs. However, Powers *et al.* [54] and Blok *et al.* [55] found that the ITS1-5.8S-ITS2 sequences of *M. incognita*, *M. javanica*, and *M. arenaria* were extremely conserved. Landa *et al.* (2008) [56] also found that the 18S sequences of rDNA from *M. hispanica* and *M. ethiopica* were very similar. Although *M. hispanica* and *M. ethiopica* can be clearly differentiated by their D2D3 28S ribosomal DNA sequences [56], *M. incognita*, *M. javanica*, and *M. arenaria* cannot [5]. Therefore, identification of some RKN species with rDNA-PCR technology remains difficult. The mitochondrial genome (mtDNA) provides a rich source of genetic markers for species identification [57], including parasitic nematode identification, because of their high mutation rates and maternal inheritance [58], mtDNA-PCR technology is an alternative method for precise identification of *Meloidogyne* species, to study intraspecific

variability and to follow maternal lineages [59]. The mitochondrial genes coxI and coxII have been widely used as DNA markers in various large organismal groups in the animal kingdom [60]. In terms of resolution, coxI is more capable of discriminating between species than either of the rRNA genes [61]. Sequence characterized amplified region (SCAR) markers have proven to be a very sensitive and reliable tool for the identification of RKNs and to provide an easy and rapid assessment of a large number of samples by a simple visual evaluation of gels [62]. The SCAR-PCR technique is more sensitive than other existing molecular techniques [63], providing a rapid species identification approach for turfgrass RKNs independent of morphology [64]. An increasing number of species-specific primers are being designed for the identification of RKNs [65–67]. In the present research, the species-specific primer pair Mv-F/Mv-R was designed based on the sequence of the rDNA ITS1-5.8S-ITS2 fragment in *M. vitis* sp. nov. so that it could provide a simple and rapid method for identifying this new species.

In this research, PCR amplification of ITS1-5.8S-ITS2 rDNA, D2D3 28S rDNA, and mtDNA (coxI and coxII) was used to identify RKNs. The nematode collected from grape was different from previously described RKNs. The ITS1-5.8S-ITS2 and D2D3 sequences of rDNA and the coxII sequence of mtDNA were compared with the corresponding fragments in RKNs available from GenBank. The most similar species was *M. mali*, with similarities of 87%, 93%, and 81%, respectively. The coxI mtDNA sequence was most similar to that of *M. ichinohei*, with a similarity of 85%. The phylogenetic tree based on ITS1-5.8S-ITS2 rDNA, D2D3 28S rDNA, and mtDNA (coxI and coxII) sequences showed that the new species was closely related to *M. mali* and clearly distinguished from other RKNs.

In summary, both the morphological and molecular characteristics revealed that the new RKN from grape is sufficiently different from the RKNs described to date to be considered a new RKN species. Thus, this new species was named *M. vitis* sp. nov. according to its host resource. *Meloidogyne vitis* sp. nov. and *M. mali* are similar in morphology and have a close molecular relationship; they may have evolved from the same ancestral species. Almost all grape roots were infected by RKNs in the vineyard investigated in this study, and the aged roots were decayed and necrotic due to RKN infection, which indicated that the RKNs in this vineyard had existed there for many years. However, their origin is still unknown, and they may have been introduced from outside. The phylogenetic tree based on various fragments shows that the new species has independent evolutionary trends; it is either indigenous in some regions of the world as an ancient species or has recently evolved and been widely spread by agriculture. High-density RKNs undoubtedly pose a serious threat to grape production. This species may be restricted to Luliang County, Yunnan Province, and will seriously damage grapes there by causing severe root knots, dwarfed plants and reduced fruit production. More surveys are needed to clarify the distribution of *M. vitis* sp. nov., and further research will also be necessary to determine its host range, pathogenesis, and control strategies.

## Supporting information

**S1 Fig. Sequence alignment of *Meloidogyne vitis* sp. nov. and *Meloidogyne mali* identified in this research based on ITS1-5.8S-ITS2 sequences.** (1 = *Meloidogyne vitis* sp. nov., 2 = *Meloidogyne mali*).
(TIF)

**S2 Fig. Sequence alignment of *Meloidogyne vitis* sp. nov. identified in this research and *Meloidogyne mali* from GenBank (KX430177.1) based on D2D3 sequences of 28S rDNA.** (1 = *Meloidogyne vitis* sp. nov., 2 = *Meloidogyne mali*).
(TIF)

**S3 Fig. Sequence alignment of *Meloidogyne vitis* sp. nov. and *Meloidogyne mali* identified in this research based on coxI-rRNA genes sequences.** (1 = *Meloidogyne vitis* sp. nov., 2 = *Meloidogyne mali*).
(TIF)

**S4 Fig. Sequence alignment of *Meloidogyne vitis* sp. nov. identified in this research and *Meloidogyne mali* from GenBank (KC112913.1) based on coxII-16S rRNA genes sequences.** (1 = *Meloidogyne vitis* sp. nov., 2 = *Meloidogyne mali*).
(TIF)

**S1 Raw images.**
(PDF)

## Author Contributions

**Conceptualization:** Yanmei Yang, Xianqi Hu, Li Chen.

**Data curation:** Yanmei Yang, Xianqi Hu, Pei Liu.

**Formal analysis:** Yanmei Yang, Xianqi Hu, Pei Liu.

**Funding acquisition:** Xianqi Hu.

**Investigation:** Yanmei Yang, Qi Zhang.

**Methodology:** Yanmei Yang, Xianqi Hu, Pei Liu, Huan Peng, Qiaomei Wang.

**Resources:** Huan Peng.

**Software:** Yanmei Yang, Pei Liu, Qiaomei Wang.

**Supervision:** Xianqi Hu.

**Writing – original draft:** Yanmei Yang.

**Writing – review & editing:** Xianqi Hu, Pei Liu.

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
