## [Decision Letter · Decision Letter 0]

2 Sep 2020

PONE-D-20-14453

A new root-knot nematode, Meloidogyne vitis sp. nov. (Nematoda: Meloidogynidae), parasitizing grape in Yunnan

PLOS ONE

Dear Dr. Hu,

Thank you for submitting your manuscript to PLOS ONE. After careful consideration, we feel that it has merit but does not fully meet PLOS ONE’s publication criteria as it currently stands. Therefore, we invite you to submit a revised version of the manuscript that addresses the points raised during the review process. The points raised are so numerous that I feel that I need to ask for a revision that deals with reviewers' comments.  I do sense that this manuscript will be an important contribution to the literature once all issues are taken care of.

We look forward to receiving your revised manuscript.

Kind regards,

Ulrich Melcher

Academic Editor

PLOS ONE

Journal Requirements:

Reviewers' comments:

Reviewer's Responses to Questions

**Comments to the Author**

1. Is the manuscript technically sound, and do the data support the conclusions?

Reviewer #1: Yes

Reviewer #2: Yes

2. Has the statistical analysis been performed appropriately and rigorously? 

Reviewer #1: N/A

Reviewer #2: Yes

3. Have the authors made all data underlying the findings in their manuscript fully available?

Reviewer #1: Yes

Reviewer #2: Yes

4. Is the manuscript presented in an intelligible fashion and written in standard English?

Reviewer #1: Yes

Reviewer #2: Yes

5. Review Comments to the Author

Reviewer #1: The authors need to work on better presenting/describing the Materials and Methods (M&M) used. Too many areas lack depth of detail for replication and much of the M&M is presented in the Results section, especially the phylogenetic data. Look at lines 499-501, none of this is in the M&M section, all of that needs to be presented in the correct section. Also, (for example) do not start sentences with M. incognita, it is Meloidogyne incognita. I think the molecular identification is the strength of this paper. The de Man indices (not formula) are helpful but limited but I can appreciate the effort there.

Line comment

52 cultivation has been conducted for thousands

58 Root-knot nematode infestation of grape has been documented in southern

59 “fleshy”?, not really.

60-61 only roots are infected, so not first, infection does not expand knots are induced

67 double citation Liu et al. (2018) [18], same with line 73

77 to further increase damage to grapes

87-97 remove, this is for materials and methods, and results sections

137 much more information about scanning electron microscope is needed.

140 Isozyme needs more detail

207 severely nematode

212 Adult female heads were found associated with the xylem

356 DGO is used several times define, I am assuming this is the DEGO, dorsal esophageal gland opening.

408 define abbreviations

426-425 remove these are M&M

456 not in the M&M sections

577-579 Is it? I haven’t done it in 30 years

625-633 repeating of M&M and results

Reviewer #2: This is a well prepared manuscript presenting new and significant information on "A new root-knot nematode, Meloidogyne vitis sp. nov. (Nematoda:Meloidogynidae), parasitizing grape in Yunnan. The species appears to be new which is well characterized by both morphological and molecular means as well as differentiated from its closely related species. The photomicrographs, SEM images and some of the line drawings are of excellent quality precisely showing the structures of particular value and interest and as given in the text. The manuscript is recommended for publication in PLOS ONE.

6. PLOS authors have the option to publish the peer review history of their article (what does this mean?). If published, this will include your full peer review and any attached files.

Reviewer #1: No

Reviewer #2: **Yes: **Zafar Ahmad Handoo

---

## [Author Response · Author response to Decision Letter 0]

25 Sep 2020

Dear editors and reviewers:

Thank you very much for reviewing our manuscript and giving us so much valuable guidance, We have modified all the problems pointed out in our manuscript, the details as follows:

Response to the Reviewer #1:

The main problems pointed out by reviewer #1 and my corresponding modification as follows:

1. Look at lines 499-501, none of this is in the M&M section, all of that needs to be presented in the correct section.

Answer: The method covered in lines 499-501 about the methods of outgroup taxa chosen and Phylogenetic trees construction has been removed and was stated in the materials and methods section. Besides, the same problem exists in Lines 450-451, 473-474, 522-523, all of these have been removed.

2. Do not start sentences with M. incognita, it is Meloidogyne incognita.

Answer: Lines 67 and 70 of manuscript both start sentences with “M. incognita”, and the “M. incognita” have been replaced with “Meloidogyne incognita”. Besides, the same problem exists in Lines 68, 338, 353, 358, 366, 369, 380, 387, 393, 397, 401, 455, 476, 478, 502, 504, 524, 636, they all have been modified. 

3. de Man indices (not formula)

Answer: The word “formula” in the line 124 has been modified as “indices”.

The following is the response to line comment of Reviewer #1

4. Line 52: cultivation has been conducted for thousands

Answer: The sentence “cultivation has a history of thousands of years” in the line 52 has been modified as “cultivation has been conducted for thousands” .

5. Line 58: Root-knot nematode infestation of grape has been documented in southern

Answer: The sentence “Grape RKN disease has been discovered in southern” in the line 58 has been modified as “Root-knot nematode infestation of grape has been documented in southern”. 

6. Line 59: “fleshy”?, not really.

Answer: The “peculiar fleshy roots are” in the line 59 has been modified as “root system are well developed and is”.

7. Lines 60-61: only roots are infected, so not first, infection does not expand knots are induced

Answer: After consideration, the sentence “The roots of grapes are usually infected first, and the infection expands to form a root knot” in lines 60-61 has been removed.

8. Line 67: double citation Liu et al. (2018) [18], same with line 73

Answer: The “[18]” contained in the sentence “Liu et al. (2017) [18] reported that grapes from the Huaihai economic zone were infected by M. incognita” of line 67 was moved to the end of the sentence. Same with line 73.

9. Line 77: to further increase damage to grapes

Answer: The “to destroy grapes” in the line 77 has been modified as “to further increase damage to grapes”.

10. Lines: 87-97 remove, this is for materials and methods, and results sections

Answer: the contents in lines 87-97 has been removed.

11. Line 137: much more information about scanning electron microscope is needed.

Answer: The instrument type and manufacturer of scanning electron microscope were added in the line 137.

12. Line 140: Isozyme needs more detail

Answer: Much more information about isozyme were added, mainly include sample preparation, electrophoretic procedure, gel stain, and a reference was added and listed in the reference section. Therefore, the order in which the reference appear in manuscript text and reference section were realignment.

13. Line 207: severely nematode

Answer: The “seriously nematode” in the line 207 has been modified as “severely nematode”.

14. Line 212: Adult female heads were found associated with the xylem

Answer: The sentence “The female adults could be seen biting tightly on the xylem when the cuticle was removed” in lines 212-213 has been modified as “Adult female heads were found associated with the xylem”.

15. Line 356: DGO is used several times define, I am assuming this is the DEGO, dorsal esophageal gland opening.

Answer: All the word “DGO” in this manuscript has been modified as “DEGO”.

16. Line 408: define abbreviations

Answer: The abbreviation “Mdh” and “Est” has been defined in “Isozyme phenotype electrophoresis” section of materials and methods.

17. Lines 426-425: remove these are M&M

Answer: After carefully reading, the contents of lines 423-425 (not 426-425) are M&M, so, the contents of lines 423-425 were removed and the following statements were slightly adjusted.

18. Line 456: not in the M&M sections

Answer: The sequence alignment method has been added to “Phylogenetic analyses” section of materials and methods, and slightly adjusted the related content; Besides, the same problem exists in lines 479, 505, 526, so, they were slightly adjusted as the same the line 456. 

19. Lines 577-579: Is it? I haven’t done it in 30 years

Answer: The contents of lines 577-579 has been removed and the reference [48], [49] also have been removed, and slightly added some related contents. Therefore, the order in which the reference appear in manuscript text and reference section were realignment.

20. Lines 625-633: repeating of M&M and results

Answer: The contents of lines 625-633 has been removed.

Response to Napsi Szincsak:

1. In the Methods section, include a sub-section called "Nomenclature".

Answer: Root-knot nematode belong to invertebrate animal, therefore, the nomenclatural acts in my manuscript was carried out according to the guidelines of zoological names in Plos One website: http://www.plosone.org/static/guidelines#botanical. The detail content of nomenclatural acts was added into our manuscript.

2. In the Results section, indicate where the globally unique identifier (GUID) will be added using XXXXX or an equivalent placeholder.

Answer: We have indicated where the globally unique identifier (GUID) would be added using “XXXXXXX”.

Besides, How do I obtain the globally unique identifier (GUID) for the new species？I need your help, thank you very much！

Response to Ulrich Melcher:

Answer: We have main modified the reference of our manuscript according to PLOS ONE's style requirements in website:

The main modifications are as follows: (a) the journal name was abbreviated. (b) the references [20] was cited as chapter in a book, so, it was modified according to the guidelines for chapter in a book references of PLOS ONE's style requirements. (c) use the symbol “[ ]” to enclose the title of chinese reference. (d) the references [25], [26], [51] was cited as book, so, it was modified according to the guidelines for book references of PLOS ONE's style requirements. (e) the references [39] was cited as Masters' theses, so, it was modified according to the guidelines for Masters' theses references of PLOS ONE's style requirements.

2. The problem about data Availability statement

Answer: All sequences are available from the GenBank database (accession number MN816222, MN816223, MN816224, MN816225, MN816226, MN814829, MN814830, MN814831, MT012386 and MT012387). Other relevant data are within the manuscript and its Supporting Information files.

3. The problem about original uncropped and unadjusted images

Answer: The original uncropped and unadjusted images of our manuscript have been submitted as supporting information with PDF file, the file name is 'S1_raw_images'.

4. PLOS requires an ORCID iD for the corresponding author

Answer: I have registered an ORCID iD and updated my Information in PLOS ONE editorial manager submission system. 

5. The problem that using the Preflight Analysis and Conversion Engine (PACE) digital diagnostic tool to deal with image.

Answer: The figure files have been modified using the Preflight Analysis and Conversion Engine (PACE) digital diagnostic tool except the Supporting Information files.

We appreaited the valuable comments from the reviewers and the editors. Thank you very much for your continued attention.

Sincerely

Xianqi Hu

Professor

College of Plant Protection

Yunnan Agricultural University

Kunming 650201, Yunnan, China

xqh@ynau.edu.cn

---

## [Editor Report · Decision Letter 1]

29 Sep 2020

PONE-D-20-14453R1

A new root-knot nematode, Meloidogyne vitis sp. nov. (Nematoda: Meloidogynidae), parasitizing grape in Yunnan

PLOS ONE

Dear Dr. Hu,

Thank you for submitting your manuscript to PLOS ONE. After careful consideration, we feel that it has merit but does not fully meet PLOS ONE’s publication criteria as it currently stands. Therefore, we invite you to submit a revised version of the manuscript that addresses the points raised during the review process.

We look forward to receiving your revised manuscript.

Kind regards,

Ulrich Melcher

Academic Editor

PLOS ONE

Additional Editor Comments (if provided):

This revised manuscript is mostly well written and reports descriptors for the proposed new nematode species. I would recommend acceptance except for the observation that the abstract is poorly written. There many errors in Engish usage. They need to ce corrected before publication. The authors have shown that they are able to act on such suggestions.

The corrections I could find wee:

In absTRACT:

…pore is located on ventrally region pore is located on the ventral region

I don’t understand the difference between “not smooth{ and :slightly wrinkled”

…The new species has rare Mdh (N3d) and Est phenotypes (VF1).

Also the following needs work for clarity.

“Collected M. mali and amplified the sequences

as mentioned above and compared with the corresponding sequence of new species,

the result showed that all of these sequences were significantly different.”

Different from what?

Second-stage

juveniles are characterized by a head region that is not smooth and slightly wrinkled, a

labial disc fused with the medial lips to form a dumbbell-shaped structure, a slightly

sunken into the middle of the medial lips

The new species has rare Mdh and Est phenotypes (VF1)

SCAR?

---

## [Author Response · Author response to Decision Letter 1]

26 Oct 2020

Dear editors and reviewers:

Thank you very much for reviewing our manuscript and giving us so much valuable guidance, we have modified all the problems pointed out in our manuscript, the details as follows:

Response to the Additional Editor:

The main problems pointed out by additional editor and my corresponding modification as follows: 

In abstract：

1. …pore is located on ventrally region pore is located on the ventral region

Answer: The sentence “pore is located on ventrally region” in the line 21 has been modified as “pore is located on the ventral region”.

2. I don’t understand the difference between “not smooth{ and :slightly wrinkled”

Answer: The words “not smooth” and “slightly wrinkled” in the sentence “Second-stage juveniles are characterized by a head region that is not smooth and slightly wrinkled” both mean the head region is not smooth and they have no obvious difference. After consideration, the sentence “Second-stage juveniles are characterized by a head region that is not smooth and slightly wrinkled” in lines 27-28 has been modified as “Second-stage juveniles are characterized by a head region with slightly wrinkled mark”.

3. …The new species has rare Mdh (N3d) and Est phenotypes (VF1).

Answer: The sentence “The new species has a rare Mdh phenotype (N3d) and Est phenotype (VF1)” in the line 31 has been modified as “The new species has rare Mdh (N3d) and Est phenotypes (VF1)” .

4. “Collected M. mali and amplified the sequences as mentioned above and compared with the corresponding sequence of new species, the result showed that all of these sequences were significantly different.” Different from what?

Answer: The sentence “Collected M.mali and amplified the sequences as mentioned above and compared with the corresponding sequence of new species, the result showed that all of these sequences were significantly different” in lines 34-36 is want to express that there are highly base divergence between the new species and M.mali in the sequences as mentioned above, therefore, this sentence has been modified as “Meloidogyne mali was collected for amplifying these sequences as mentioned above, which were compared with the corresponding sequences of new species, the result showed that all of these sequences with highly base divergence (48-210 base divergence)” .

5. Second-stage juveniles are characterized by a head region that is not smooth and slightly wrinkled, a labial disc fused with the medial lips to form a dumbbell-shaped structure, a slightly sunken into the middle of the medial lips

Answer: The sentence “Second-stage juveniles are characterized by a head region that is not smooth and slightly wrinkled, a labial disc fused with the medial lips to form a dumbbell-shaped structure, a slightly sunken into the middle of the medial lips” in lines 27-29 has been modified as “Second-stage juveniles are characterized by a head region with slightly wrinkled mark, a labial disc fused with the medial lips to form a dumbbell-shaped structure, a slightly sunken appearance of the middle of the medial lips”. 

6. SCAR?

Answer: The word “SCAR” is the abbreviations of “sequence characterized amplified region”. Specific primers have been developed to PCR-amplify diagnostic repetitive regions of sequence: sequence characterized amplified region (SCAR). SCAR primers are designed based on random amplified polymorphic DNA (RAPD) product and rDNA sequences and are preferred for identification purposes as the relatively high annealing temperatures that are used with species-specific primers enhance their specificity. After consideration, the sentence “SCAR primers were designed” has been modified as “sequence characterized amplified region (SCAR) primers for rapid identification of this new species were designed”.

Response to Ulrich Melcher:

1. The problem about financial disclosure

Answer: The financial disclosure of our manuscript do not make changes. 

2. The problem about laboratory protocols

Answer: We have no laboratory protocols.

3. The problem that using the Preflight Analysis and Conversion Engine (PACE) digital diagnostic tool to deal with image.

Answer: The figure files have been modified using the Preflight Analysis and Conversion Engine (PACE) digital diagnostic tool except the Supporting Information files.

We appreaited the valuable comments from the reviewers and the editors. Thank you very much for your continued attention.

Sincerely

Xianqi Hu

Professor

College of Plant Protection

Yunnan Agricultural University

Kunming 650201, Yunnan, China

xqh@ynau.edu.cn

---

## [Editor Report · Decision Letter 2]

26 Dec 2020

A new root-knot nematode, Meloidogyne vitis sp. nov. (Nematoda: Meloidogynidae), parasitizing grape in Yunnan

PONE-D-20-14453R2

Dear Prof. Xianqi Hu

We’re pleased to inform you that your manuscript has been judged scientifically suitable for publication and will be formally accepted for publication once it meets all outstanding technical requirements.

Kind regards,

Ebrahim Shokoohi

Academic Editor

PLOS ONE

Additional Editor Comments (optional):

Dear Prof Xianqi Hu

I am pleased to inform you that you and your colleague's paper has been accepted for publication in PLOS ONE. The data provided by your team, two times revision by the colleagues, and sufficient information on the molecular identification of the new species of this problematic genus of nematode along with the other relevant information on the morphology of different stage of the new species of Meloidogyne make this paper suitable for publishing in PLOS ONE. Additionally, I have checked all the comments of twice revision which have been implemented in the manuscript. Therefore, I have decided to accept your manuscript for PLOS ONE. However, you have to follow the journal style and check all the citations within the text and in the reference part to be in accordance.

With kind regards,

Ebrahim

Reviewers' comments:

No comment

---

## [Editor Report · Acceptance letter]

6 Jan 2021

PONE-D-20-14453R2 

A new root-knot nematode, *Meloidogyne vitis* sp. nov. (Nematoda: Meloidogynidae), parasitizing grape in Yunnan 

Dear Dr. Hu:

I'm pleased to inform you that your manuscript has been deemed suitable for publication in PLOS ONE. Congratulations! Your manuscript is now with our production department. 

Kind regards, 

on behalf of

Dr. Ebrahim Shokoohi 

Academic Editor

PLOS ONE